# Mechanical loading reveals an intrinsic cardiomyocyte stiffness contribution to diastolic dysfunction in murine cardiometabolic disease

Johannes V. Janssens[1,2], Antonia J. A. Raaijmakers[1], Parisa Koutsifeli[3], Kate L. Weeks[1,4] [ID], James R. Bell[1,5], Jennifer E. Van Eyk[2], Claire L. Curl[1], Kimberley M. Mellor[1,3,6] [ID] and Lea M. D. Delbridge[1] [ID]

[1]*Department of Anatomy & Physiology, University of Melbourne, Melbourne, Australia*
[2]*Smidt Heart Institute, Cedars-Sinai Medical Center, Los Angeles, CA, USA*
[3]*Auckland Bioengineering Institute, University of Auckland, New Zealand*
[4]*Baker Department of Cardiometabolic Health (Baker), University of Melbourne, Melbourne, Australia*
[5]*Department of Microbiology, Anatomy, Physiology & Pharmacology, La Trobe University, Melbourne, Australia*
[6]*Department of Physiology, University of Auckland, New Zealand*

The peer review history is available in the Supporting Information section of this article (https://doi.org/10.1113/JP286437#support-information-section).

*The Journal of Physiology*

**Abstract figure legend** Understanding cardiomyocyte stiffness components is an important priority for identifying new therapeutics for diastolic dysfunction, a key feature of cardiometabolic disease. In this study cardiac function was measured *in vivo* (echocardiography) for mice fed a high-fat/sugar diet (HFSD, ≥25 weeks). Performance of intact isolated cardiomyocytes derived from the same hearts was measured during pacing under non-loaded, loaded and stretched conditions *in vitro*. Calibrated cardiomyocyte stretches demonstrated that stiffness was elevated in HFSD cardiomyocytes *in vitro* and correlated with diastolic dysfunction (*E/e′*) *in vivo*. These findings show that stiff hearts are characterized by stiff cardiomyocytes in metabolic disease.

This article was first published as a preprint. Janssens JV, Raaijmakers AJA, Koutsifeli P, Weeks KL, Bell JR, Van Eyk JE, Curl CL, Mellor KM, Delbridge LMD. 2024. Mechanical loading reveals an intrinsic cardiomyocyte stiffness contribution to diastolic dysfunction in murine cardiometabolic disease. bioRxiv. https://doi.org/10.1101/2024.02.21.581448

**Abstract**  Cardiometabolic syndromes including diabetes and obesity are associated with occurrence of heart failure with diastolic dysfunction. There are no specific treatments for diastolic dysfunction, and therapies to manage symptoms have limited efficacy. Understanding of the cardiomyocyte origins of diastolic dysfunction is an important priority to identify new therapeutics. The investigative goal was to experimentally define *in vitro* stiffness properties of isolated cardiomyocytes derived from rodent hearts exhibiting diastolic dysfunction *in vivo* in response to dietary induction of cardiometabolic disease. Male mice fed a high fat/sugar diet (HFSD *vs.* control) exhibited diastolic dysfunction (echo $E/e'$ Doppler ratio). Intact paced cardiomyocytes were functionally investigated in three conditions: non-loaded, loaded and stretched. Mean stiffness of HFSD cardiomyocytes was 70% higher than control. $E/e'$ for the HFSD hearts was elevated by 35%. A significant relationship was identified between *in vitro* cardiomyocyte stiffness and *in vivo* dysfunction severity. With conversion from the non-loaded to loaded condition, the decrement in maximal sarcomere lengthening rate was more accentuated in HFSD cardiomyocytes (*vs.* control). With stretch, the $Ca^{2+}$ transient decay time course was prolonged. With increased pacing, cardiomyocyte stiffness was elevated, yet diastolic $Ca^{2+}$ elevation was attenuated. Our findings show unequivocally that cardiomyocyte mechanical dysfunction cannot be detected by analysis of non-loaded shortening. Collectively, these findings demonstrate that a component of cardiac diastolic dysfunction in cardiometabolic disease is derived from cardiomyocyte stiffness. Differential responses to load, stretch and pacing suggest that a previously undescribed alteration in myofilament–$Ca^{2+}$ interaction contributes to intrinsic cardiomyocyte stiffness in cardiometabolic disease.

(Received 18 April 2024; accepted after revision 4 November 2024; first published online 26 November 2024)
**Corresponding author** L. M. D. Delbridge: Cardiac Phenomics Laboratory, Department of Anatomy and Physiology, University of Melbourne, Grattan Street, Parkville, Victoria, Australia, 3010.    Email: lmd@unimelb.edu.au

## Key points

- Understanding cardiomyocyte stiffness components is an important priority for identifying new therapeutics for diastolic dysfunction, a key feature of cardiometabolic disease.
- In this study cardiac function was measured *in vivo* (echocardiography) for mice fed a high-fat/sugar diet (HFSD, ≥25 weeks). Performance of intact isolated cardiomyocytes derived from the same hearts was measured during pacing under non-loaded, loaded and stretched conditions *in vitro*.
- Calibrated cardiomyocyte stretches demonstrated that stiffness (stress/strain) was elevated in HFSD cardiomyocytes *in vitro* and correlated with diastolic dysfunction ($E/e'$) *in vivo*. HFSD cardiomyocyte $Ca^{2+}$ transient decay was prolonged in response to stretch. Stiffness was accentuated with pacing increase while the elevation in diastolic $Ca^{2+}$ was attenuated.
- Data show unequivocally that cardiomyocyte mechanical dysfunction cannot be detected by analysis of non-loaded shortening.
- These findings suggest that stretch-dependent augmentation of the myofilament–$Ca^{2+}$ response during diastole partially underlies elevated cardiomyocyte stiffness and diastolic dysfunction of hearts of animals with cardiometabolic disease.

**Johannes Janssens** completed his PhD in the Department of Anatomy and Physiology at the University of Melbourne (Australia) where the current work was performed. After completing his PhD he received a Fulbright Postdoctoral Scholarship to develop a novel blood biomarker of diabetic heart disease at Cedars-Sinai Medical Centre (Los Angeles, CA) where he continues to work currently. Johannes' research aspirations involve linking large-scale proteomic features of hearts and cardiomyocytes with their functional phenotype to develop novel therapeutic and diagnostic strategies for cardiac pathophysiology. In particular diseases where diastolic dysfunction is a prominent characteristic.

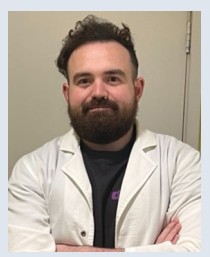

## Introduction

Increasingly, cardiometabolic syndromes, including diabetes and obesity, are associated with occurrence of heart failure with a diastolic dysfunction patho-phenotype (Sotomi et al., 2021). Diastolic dysfunction is characteristic of heart failure with preserved ejection fraction (HFpEF), which has now overtaken heart failure with reduced ejection fraction (HFrEF) in comprising the majority of new heart failure diagnoses (Jackson et al., 2022; McHugh et al., 2019). Diastolic dysfunction is also linked with progression of diabetic cardiomyopathy (Patil et al., 2011; Redfield, 2016). With both HFpEF and diabetic heart disease, early development of diastolic dysfunction can be detected even when other phenotypic features/comorbidities are heterogeneous (i.e. hypertrophy, dilatation, natriuretic factor elevation, hypertension) (Kittleson et al., 2023). Common clinical features of cardiometabolic syndromes, including obesity and insulin resistance, are independent risk factors for diastolic dysfunction as a prelude to heart failure (Kamon et al., 2020; Rozenbaum et al., 2019). The experimental induction of these phenotypes using a high fat/sugar diet feeding protocol has been shown by us and others to reliably produce diastolic dysfunction in rodents (Daniels et al., 2022; Dia et al., 2020; Pulinilkunnil et al., 2014; Sowers et al., 2020; Wingard et al., 2021). Cardiometabolic syndromes can involve the 'silent' presence and progression of diastolic dysfunction for an extended period. Symptoms can be subclinical and/or detectable only in challenge conditions (i.e. as reduced diastolic reserve in exercise) or when exacerbated by co-morbidity (Lam et al., 2019; Leung et al., 2015). Diagnosis of heart failure with diastolic dysfunction is independently associated with higher incidence of adverse clinical events (Sotomi et al., 2021). Notably, there is currently no specific treatment for diastolic dysfunction and/or HFpEF, and therapies to manage symptoms have limited efficacy (Delbridge et al., 2022).

In diastolic dysfunction, the relaxation and filling phases of the cardiac cycle are impaired. The condition is generally described qualitatively as an increase in left ventricular wall 'stiffness' (or by a decrease in compliance, the inverse of stiffness) (Daniels et al., 2022; Janssens et al., 2024). Clinically and experimentally, reduced compliance can be measured invasively as an elevation in left ventricular filling pressure or an increase in the gradient of the diastolic pressure–volume relationship. Non-invasively, on Doppler echocardiogram, an increase in the $E/e'$ ratio (early ($E$) mitral valve blood flow rate to early mitral annulus tissue motion ($e'$)) is an index used to identify diastolic abnormality (Lo & Thomas, 2010).

Myocardial stiffness is commonly considered to reflect myocardial extracellular matrix abnormalities linked with increased tissue fibrosis. While a component of reduced compliance likely reflects fibrotic infiltration, diastolic dysfunction can be identified in settings where there is no evidence of increased interstitial fibrosis burden (Curl et al., 2018). There is growing appreciation that alterations of myocardial compliance in diastolic dysfunction disease states represents underlying intrinsic cardiomyocyte stiffness pathology which contributes to impaired relaxation and filling (Villalobos Lizardi et al., 2022; Zhou & Pu, 2016). Understanding the cardiomyocyte origin of diastolic dysfunction is an important priority to identify new molecular targets and to support the development of new diagnostics and therapeutics to reduce morbidity and mortality in cardiometabolic disease states.

Cardiomyocyte origins of diastolic dysfunction potentially comprise structural and operational components. Evidence of molecular changes in sarcomeric and cytoskeletal elements is extensive (Janssens et al., 2024; Loescher et al., 2023; Villalobos Lizardi et al., 2022). Operationally, the cardiac cycle of contraction and relaxation is driven by the cyclic flux of cardiomyocyte activator $Ca^{2+}$ to initiate contraction (myocyte sarcomere shortening) and to allow relaxation (myocyte sarcomere lengthening). The relationship between cytosolic $Ca^{2+}$ levels and myofilament activation is dynamic and depends on several external factors some of which include stretch and pacing (Fabiato & Fabiato, 1978; Varian & Janssen, 2007).

Conventionally, the steady-state force–$Ca^{2+}$ analysis (typically performed in permeabilized, quiescent preparations) is used to summarise the myofilament–$Ca^{2+}$ response and is the product of many complex myofilament protein interactions. Less work has been done to investigate the dynamic changes in the myofilament–$Ca^{2+}$ relationship in paced intact cardiomyocytes, especially in disease contexts. In diastolic dysfunction, specifically in HFpEF animal models, evidence is accruing that $Ca^{2+}$ deficit and impaired non-loaded shortening *are not* phenotypic (Kilfoil et al., 2020). Indeed, cardiomyocyte myofilament hyper-activation and elevated $Ca^{2+}$ levels have been reported (Curl et al., 2018; Louch, 2020; Saad et al., 2023). As we have recently proposed, the role for elevated diastolic $Ca^{2+}$ levels and augmented myofilament–$Ca^{2+}$ response comprising a cross-bridge-derived component of cardiomyocyte diastolic stiffness is increasingly plausible (Janssens et al., 2024). This experimental enquiry focuses on intact cardiomyocyte mechanical and $Ca^{2+}$ performance in response to load, stretch and modulated pacing conditions.

While the term 'stiffness' is used as a general descriptor of impaired compliance in a cardiovascular setting, it is more formally specified quantitatively in physical

science disciplines (Mirsky, 1976). Stiffness is defined as the extent to which an object resists deformation. Stiffness is influenced by the geometry and the intrinsic material properties of an object. In the cardiac literature and in this study, stress/strain relations are calculated to assess cardiomyocyte resistance to stretch. Stress is defined as force per cross-sectional area, and strain is the application of a normalized stretch or deformation. The ratio of stress/strain defines the cardiomyocyte resistance to stretch independent of geometry and is known as Young's modulus. While stiffness and Young's modulus are not equivalent, the term 'stiffness' is commonly used (even though Young's modulus is the correct formal term). In the cardiac setting, a stretch or deformation state is commonly described as 'loading' – for example 'pre-load' (diastolic stretch) and 'after-load' (resistance to shortening). Depending on the experimental design, stiffness behaviour of myocardial tissues, active cardiomyocytes and/or myocyte subcellular cytoskeleton and sarcomere structures all potentially contribute to *in vivo* functional stiffness. Little is known about how stiffness measured at different structural levels may determine function *in vivo* and how stiffness elements are quantitatively changed in pathophysiology – in particular in acquired disease conditions. Thus, the goal of this investigation was to experimentally characterize *in vitro* stiffness properties (stress/strain) of cardiomyocytes derived from hearts of animals exhibiting diastolic dysfunction *in vivo* in the setting of dietary induction of cardiometabolic disease.

## Methods

### Ethical approval

The authors understand the ethical principles under which *The Journal of Physiology* operates, and the work presented in this study complies with the animal ethics checklist as outlined in Instructions to Authors. All experiments conform to the principles and regulations as described by Grundy (2015). Animals (male C57Bl/6J mice) were purchased from the Animal Resource Centre (Perth, WA, Australia). All animal experiments were approved by the University of Melbourne (School of Biomedical Sciences) Animal Ethics Committee (Project Ethics Approval 10366) and complied with the guidelines and regulations of the Code of Practice for the Care and Use of Animals for Scientific Purposes (National Health and Medical Research Council Australia). Animals were provided with free access to food and water, were group-housed in a temperature-controlled environment of 21–23°C with 12 h light–dark cycles. Animals were euthanized under anaesthesia (single dose of sodium pentobarbital 70 mg/kg I.P.) on cardiac excision.

### Animal model of cardiometabolic disease

As a pre-clinical cardiometabolic disease model, the high fat/sugar diet (HFSD) fed rodents recapitulates a common and growing human disease phenotype (obesity, insulin resistance, glucose intolerance) and has been established as a diastolic dysfunction model (Heather et al., 2022). Cardiometabolic disease was induced in male C57Bl/6J mice by a HFSD intervention commencing at 8–9 weeks of age. Animals were randomized to experimental treatment groups and were group housed (>4 animals per cage) in a temperature-controlled environment of 21–23°C with 12 h light–dark cycles. After 1 week transitional feeding period (alternating feed periods), animals were fed a HFSD (43% kcal from fat, 200 g/kg sucrose, SF04-001, Specialty Feeds, Glen Forrest, WA, Australia) or control reference diet (CTRL: 16% kcal from fat, 100 g/kg sucrose, AIN93G, Specialty Feeds, Glen Forrest, WA, Australia) for at least 25 weeks.

### Glucose tolerance testing

At 23–25 weeks of feeding, glucose tolerance testing was performed in mice following 6 h fasting as previously described (Chandramouli et al., 2018). Baseline blood glucose levels were measured using an Accu-Chek glucometer (Roche Diabetes Care, North Ryde, NSW, Australia with a blood sample obtained from a needle prick to the tail vein. Glucose (1.5 g/kg body weight) was injected I.P. and blood glucose was measured at 5, 15, 30, 60, 90 and 120 min after the glucose injection.

### Transthoracic echocardiography

At 25–26 weeks of feeding, cardiac structure, systolic and diastolic function were evaluated by transthoracic two-dimensional B- and M-mode and blood flow and tissue Doppler echocardiography (GE Vivid 9; 15 mHz i13L linear array transducer; GE Healthcare, Chicago, IL, USA) as described previously (Chandramouli et al., 2018; Curl et al., 2018). Briefly, mice were lightly anaesthetized with isoflurane (1.5%; Baxter Healthcare, NSW, Australia). Parasternal short axis view was used to measure morphological and systolic parameters (interventricular septum (IVS), left ventricular posterior wall (LVPW), left ventricular internal dimension (LVID), fractional shortening (FS), ejection fraction (EF), heart rate (HR)). Pulse wave Doppler and tissue Doppler scans were acquired from the apical four-chamber view to assess LV diastolic function parameters: velocity of early ($E$) and late ($A$) mitral inflow and early ($e'$) and late ($a'$) diastolic velocity of mitral annulus. From these compound indices of diastolic function $E/e'$, $E/A$ and $e'/a'$ were calculated. In addition, the slope of the $E$ wave was used to calculate

mitral valve deceleration time. Three consecutive cardiac cycles were sampled for each measurement.

## Cardiomyocyte preparation, structural and functional assessment

A series of protocols were implemented to phenotype cardiomyocyte and sarcomere morphology, and assess functional status in different contraction configurations and challenge settings.

**Cardiomyocyte isolation, morphology and micro-fluorimetry.** Prior to cardiomyocyte isolation animals were anaesthetized (sodium pentobarbital 70 mg/kg I.P.) and hearts excised. HFSD and CTRL left ventricular cardiomyocytes were enzymatically isolated using methods previously described (Howlett et al., 2006; Mellor et al., 2011) at 30–32 weeks' feeding. Briefly, the heart was excised, cannulated via the aorta, and perfused on a Langendorff apparatus with an oxygenated $Ca^{2+}$ free buffer at 37°C for 10 min. The buffer was composed of (in mM): 130 NaCl, 5 KCl, 0.33 $NaH_2PO_4$; 1 $MgCl_2$, 25 HEPES, 20 glucose, 3 sodium pyruvate, 1 sodium lactate. The $Ca^{2+}$ free buffer was then supplemented with 0.66 mg/ml collagenase (Worthington Biochemical Corp., Lakewood, NJ, USA; Type II), 0.033 mg/ml trypsin and 0.05 mM $CaCl_2$ and perfusion continued for a further 10 min. The heart was removed from the Langendorff apparatus, and the left ventricle was segmented into small pieces in a high potassium solution composed of (in mM): 30 KCl, 90 KOH, 30 $KH_2PO_4$, 3 $MgSO_4$, 50 glutamic acid, 20 taurine, 0.5 EGTA, 10 glucose, and 10 HEPES. Single left ventricular cardiomyocytes were isolated via gentle trituration. Viable cardiomyocytes were resuspended in HEPES–Krebs buffer (in mM: 146.2 NaCl, 4.69 KCl, 0.35 $NaH_2PO_4H_2O$, 1.05 $MgSO_47H_2O$, 10 HEPES, 11 glucose containing 1.8 mM $Ca^{2+}$) and were stored on a rocking platform at room temperature. From each heart cell, dimensions were collected from at least 50 rod shaped cardiomyocytes using an inverted light microscope (Nikon, Tokyo, Japan) and calibrated eye piece as previously described (Curl et al., 2018). Cardiomyocyte volume was calculated by inserting experimentally determined myocyte length and width values into the formula: volume = length × width × depth × multiplier 0.54 (elliptical area of the rectangle defined by width and length; Satoh et al., 1996). Cardiomyocyte width: depth ratio was assumed to be 1.44 (Natali et al., 2001). Cardiomyocyte cross-sectional area (CSA) could be calculated from measured width scaled for two-dimensional geometry (CSA = $W$ [$W$/1.44] × 0.54 μm²). An aliquot of isolated cardiomyocyte suspension was pipetted into a cell chamber mounted on the stage of an inverted fluorescence microscope (Motic AE31, Kowloon, Hong Kong) and superfused with HEPES–Krebs solution with 2 mM $Ca^{2+}$, 37°C at a flow rate of 2 mL/min. Cardiomyocyte sarcomere length and $Ca^{2+}$ transients were simultaneously recorded during isotonic contractions. Myocyte force development was evaluated in the same cells with transducer attachment, using four protocols described below.

**Characterizing cardiomyocyte diastolic function: non-loaded shortening measurements (isotonic), loaded force measurements (auxotonic), stretch responses and pacing challenge.** Cardiomyocyte performance was evaluated in several paced conditions: designated 'non-loaded' (isotonic), 'loaded' (auxotonic after-loaded), and 'stretched' (auxotonic pre-loaded). The non-loaded state refers to the absence of externally applied load (noting that sarcomeric and cytoskeletal structures do constitute an underlying internal load).

Firstly, cardiomyocyte non-loaded (isotonic) sarcomere contractions (IonOptix (Westwood, MA, USA) SarcLen, fast Fourier transform, FFT) and $Ca^{2+}$ transients (Fura2-AM, 5 μM microfluorimetry $F_{340:380nm}$ ratio) were simultaneously measured at 2 Hz, using a microscope diaphragm to define a rectangular intra-myocyte region of interest as depicted. During contraction cycles, cardiomyocyte sarcomere length was monitored within an intracellular region of interest defined by an adjustable diaphragm mounted in the microscope optical tube (Fig. 4A).

Secondly, to obtain loaded (auxotonic) contraction data, each cardiomyocyte was attached to a pair of glass fibres coated with a bio-adhesive (MyoTak, IonOptix, USA). Attachment sites were positioned to be outside the sarcomere tracking region, closer to myocyte longitudinal ends. One glass fibre was connected to a force transducer (fibre stiffness 20 N/m, IonOptix), and the other to a piezo-motor (Nano-drive, Mad City Labs Inc., Madison, WI, USA) to adjust cardiomyocyte length (Fig. 4A). With successful attachments, cardiomyocytes were lifted from the coverslip using motorized micromanipulators (MC1000E, Siskiyou Corp., Grants Pass, OR, USA) attached to both the force transducer and the piezo-motor. Baseline loaded contractions at 2 Hz were recorded. Sarcomere length changes were measured using FFT as described above. In addition, with glass fibre attachment, tracking of mean sarcomere shortening within a myocyte segment delineated by the glass fibre edge positions could be performed in parallel (Fig. 4B and C).

Thirdly, force was recorded during a standardized stretch protocol of loaded cardiomyocytes to measure the effect of increasing diastolic cardiomyocyte/sarcomere basal length during pacing. A stress–length relation was constructed for each myocyte (where stress = force/cross-sectional area). Prior to

the commencement of the myocyte stretch measurements, a protocol to calibrate the stretch step was implemented. Four consecutive 0.5 V input signals were delivered to the piezo-motor to determine a calibration constant to define the percentage cell stretch for a given voltage (Fig. 4*B*). For each myocyte a calibration constant was calculated as: myocyte segment length change per piezo-motor mV/sarcomere count in segment (where myocyte sarcomere count = myocyte segment length (μm)/mean diastolic sarcomere length (μm)). To track the glass fibre position a real-time contrast detection algorithm (IonOptix – EdgeLength) was focused on the internal edge of each of the glass fibres (Fig. 4) (Peyronnet et al., 2017). This enabled reproducible and comparable stretch steps to be applied to different cardiomyocyte preparations (Fig. 4*B* and *C*). Prior to the 'stress–length' protocol, a cell-specific calibration constant was applied to yield a standardized stretch for each cardiomyocyte evaluated. The stress–length protocol involved progressive stretch steps totalling an approximate 30% cell stretch (Fig. 4*B* and *C*).

Lastly, the pacing was increased to 4 Hz, a rate adaptation period allowed (2 min) and the stress–length protocol was repeated. Sarcomere length, force development and intracellular $Ca^{2+}$ transients were simultaneously measured. An average of at least 10 twitches or transients were used to calculate sarcomere, $Ca^{2+}$ and force parameters at a given stretch length. In summary, the following parameters were measured: diastolic stress ($mN/mm^2$), diastolic stress–length relation [$(mN/mm^2)/(\%$ length change)], diastolic stress–sarcomere length relations (($mN/mm^2$)/μm), systolic:diastolic stress–sarcomere relation (Frank–Starling Gain Index (Bollensdorff et al., 2011)), systolic: diastolic stress–length relation, systolic stress amplitude ($mN/mm^2$), sarcomere shortening (% diastolic sarcomere length (SL)), maximum sarcomere shortening and lengthening velocity (μm/s). In addition, $Ca^{2+}$ parameters were measured: diastolic $Ca^{2+}$, systolic $Ca^{2+}$ and time constant of decay of the $Ca^{2+}$ transient ($\tau$).

### Statistical analyses

Data are presented as means ± standard deviation and statistical analysis was performed using GraphPad Prism V9.5.1 (GraphPad Software, Boston, MA, USA). Cardiomyocyte stress–length and stress–SL relations were modelled with simple linear regression. For comparison between two groups Student's unpaired *t* test was used. For assessment between two groups at multiple points a one-way ANOVA with Bonferroni multiple comparisons *post hoc* test was used. Two-way ANOVA with Šídák's multiple comparisons *post hoc* test was used for the evaluation of data groups with two independent variables. Pearson correlation analyses were used to assess statistical significance of fitted linear regression lines. A *P*-value of <0.05 was considered statistically significant. Analyses were performed using a blinded protocol.

## Results

### Systemic indicators of cardiometabolic disease and *in vivo* diastolic function in HFSD mice

High fat/sugar diet-fed mice exhibited obesity and glucose intolerance. Relative to CTRL, the body weight of HFSD mice was significantly elevated within 5 weeks of dietary intervention increasing over the study duration by 50% relative to CTRL (Fig. 1*A*, CTRL: 28.1 ± 2.0 *vs.* HFSD: 44.3 ± 1.9 g). After 23–24 weeks' diet treatment, glucose tolerance testing of a subset of the HFSD cohort was performed. Glucose intolerance in response to intra-peritoneal glucose administration was evidenced by increased area under the curve, consistent with literature reporting insulin resistance in mice fed a similar diet (Fig. 1*B*).

After confirming these key cardiometabolic disease characteristics in the HFSD mice, echocardiographic functional evaluation was undertaken. Focusing on diastolic function evaluation (Fig. 1*C–G*), there was higher early mitral inflow (Fig. 1*D* and *E* wave, CTRL: 472 ± 74, HFSD: 569 ± 65 mm/s) and reduced velocity of wall motion (Fig. 1*E*, *e'* wave, CTRL: 28.5 ± 5.1, HFSD: 25.2 ± 3.5 mm/s) in HFSD myocardium. Deceleration time of mitral inflow was shortened (Fig. 1*F*, CTRL: 30.8 ± 8.7, HFSD: 19.7 ± 5.9 ms) and *E/e'* ratio increased (Fig. 1*G*, CTRL: 16.7 ± 3.9, HFSD: 22.3 ± 6.0), all observations consistent with reduced tissue compliance and diastolic dysfunction. No significant differences in systolic performance parameters were observed (Table 1), confirming the primary *in vivo* phenotype of diastolic dysfunction. The next step was to evaluate the cardio-myocyte involvement in dysfunction.

### Non-loaded sarcomere and $Ca^{2+}$ dynamics in CTRL and HFSD cardiomyocytes

Single isolated cardiomyocytes were prepared from hearts of a subset of mice which underwent echocardiographic analyses. For intact, non-loaded cardiomyocytes, sarcomere shortening and lengthening cycles were similar for CTRL and HFSD groups (Fig. 2*A*). Diastolic sarcomere length (SL) (Fig. 2*B*, CTRL: 1.71 ± 0.04, HFSD: 1.72 ± 0.04 μm) and extent of sarcomere shortening (Fig. 2*C*, CTRL: 3.79 ± 2.24, HFSD: 3.94% ± 3.15% diastolic SL) were not significantly different between groups. Examination of sarcomere contraction kinetics

(Fig. 2*D* exemplar trace) showed that the maximum rate of lengthening (Fig. 2*E*, CTRL: 1.08 ± 0.85, HFSD: 1.29 ± 1.07 μm/s) and shortening (Fig. 2*F*, CTRL: −2.32 ± 1.44, HFSD: −2.57 ± 2.01 μm/s) did not significantly differ between CTRL and HFSD groups. Collectively these findings indicate that in the non-loaded setting no differences in CTRL and HFSD cardiomyocyte systolic and diastolic non-loaded performance were detected, a finding consistent with other studies investigating non-loaded cardiomyocyte performance in diet-induced cardiometabolic disease including our previous studies in fructose-fed mice (Llano-Diez et al., 2016; Mellor et al., 2012; Wang et al., 2020; West et al., 2019).

Next, Ca$^{2+}$ parameters were compared with cardiomyocyte sarcomere lengthening performance during relaxation and diastole (Fig. 2*G*, exemplar traces). No significant difference was evident between dietary treatment groups in the systolic Ca$^{2+}$ (Fig. 2*H*, CTRL: 1.098 ± 0.169, HFSD: 1.171 ± 0.182 $F_{340:380}$), diastolic Ca$^{2+}$ (Fig. 2*I*, CTRL: 0.749 ± 0.097, HFSD: 0.761 ± 0.094 $F_{340:380}$) or time constant ($\tau$) of Ca$^{2+}$ transient decay (Fig. 2*J*, CTRL: 0.160 ± 0.067, HFSD: 0.137 ± 0.054 s). Thus, the significant diastolic dysfunction measured *in vivo* for the HFSD group was not observed in a non-loaded *in vitro* setting. To reconcile the impaired echocardiographic heart performance with absence of dysfunction in non-loaded contracting single cardiomyocytes, the next strategy was to assess single cardiomyocyte response to loading. A common feature of cardiometabolic disease is elevation in after-load – that is, elevated systemic resistance potentially impeding myocardial pump function. To simulate this *in vitro*, mechanically loaded cardiomyocyte performance was examined.

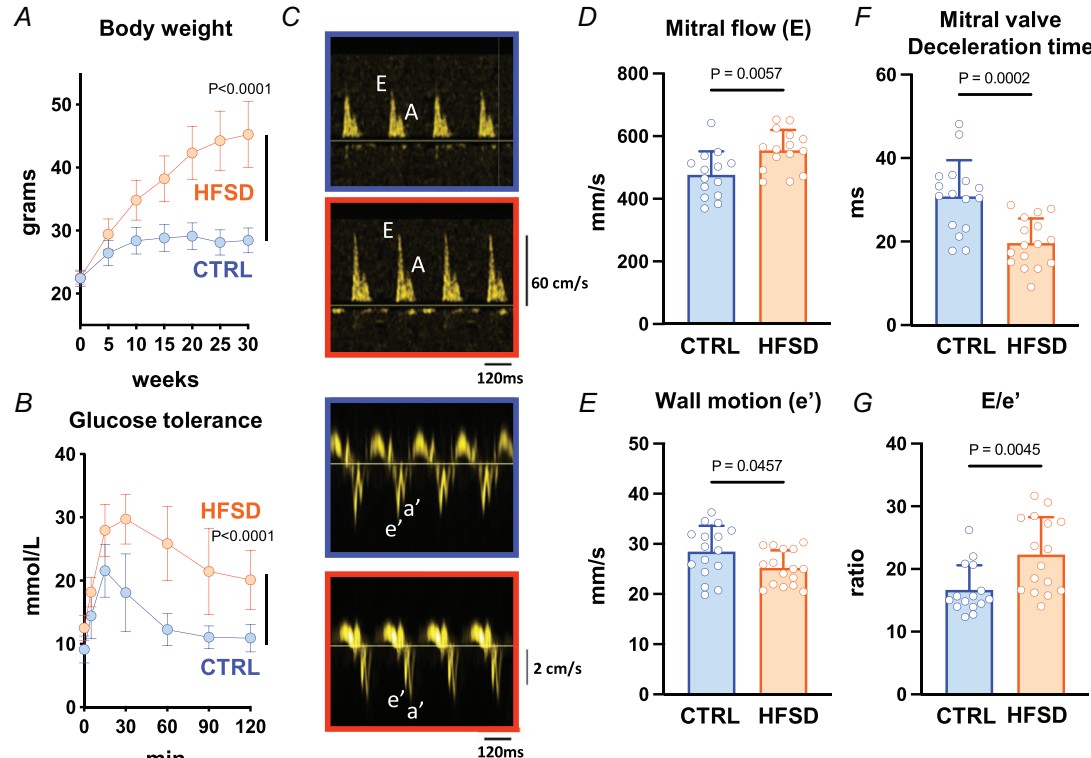

**Figure 1. Systemic indicators of cardiometabolic disease and *in vivo* diastolic function in high-fat/sugar diet (HFSD) *vs*. control (CTRL) mice**

*A*, body weight elevation at 5-week intervals up to 30 weeks of dietary treatment (last time point full cohort intact; *N* = 16–17 mice per group). *B*, glucose tolerance testing at 23–24 weeks' post dietary intervention; *N* = 6 mice per group. *C*, echocardiography exemplar blood flow and tissue Doppler images from which diastolic indices were derived for CTRL and HFSD mice. Panels with orange border: HFSD mitral blood flow and annular wall motion. Panels with blue border: CTRL mitral blood flow and annular wall motion. *N* = 14–16 mice per group. *D*, early diastolic mitral annular blood flow (*E*) was increased in HFSD mice. *E*, early diastolic mitral annular wall velocity (*e′*) was decreased in HFSD mice. *F*, deceleration time of early diastolic mitral annular blood flow was decreased in HFSD mice. *G*, ratio of early diastolic blood flow to wall velocity (*E/e′*) was increased in HFSD mice. Time-course data analysed using two-way repeated measures ANOVA with Šidák's *post hoc* test with *P*-value for 'time × diet' effect presented. Comparisons between two treatment groups performed using Student's *t* test. Data presented as means ± standard deviation. [Colour figure can be viewed at wileyonlinelibrary.com]

**Table 1. Echocardiographic parameters in CTRL and HFSD mice**

|  | CTRL | HFSD | *P* |
|---|---|---|---|
| Heart rate (bpm) | 515 ± 74 | 551 ± 29 | 0.174 |
| LV mass (mg) | 125 ± 27 | 131 ± 21 | 0.314 |
| LVIDd (mm) | 3.60 ± 0.17 | 4.16 ± 0.47 | 0.004 |
| IVSd (mm) | 0.84 ± 0.06 | 0.80 ± 0.10 | 0.349 |
| LVPWd (mm) | 0.94 ± 0.03 | 0.94 ± 0.10 | 0.830 |
| RWT (ratio) | 0.49 ± 0.06 | 0.43 ± 0.09 | 0.079 |
| Ejection fraction (%) | 67.8 ± 3.7 | 64.9 ± 3.2 | 0.093 |
| Fractional shortening (%) | 31.8 ± 3.6 | 30.6 ± 2.1 | 0.400 |
| *E* wave (mm/s) | 472 ± 74 | 569 ± 65 | 0.003 |
| *A* wave (mm/s) | 348 ± 99 | 400 ± 116 | 0.190 |
| *e′* wave (mm/s) | 28.5 ± 5.1 | 25.2 ± 3.5 | 0.046 |
| *a′* wave (mm/s) | 23.5 ± 7.8 | 23.5 ± 6.4 | 0.995 |
| *E/A* | 1.51 ± 0.41 | 1.47 ± 0.37 | 0.095 |
| *e′/a′* | 1.33 ± 0.44 | 1.12 ± 0.39 | 0.171 |
| *E/e′* | 16.7 ± 3.9 | 22.3 ± 6.0 | 0.005 |
| MV deceleration time (ms) | 30.8 ± 8.7 | 19.7 ± 5.9 | <0.001 |

Comparisons between two treatment groups performed using Students' unpaired *t* test (*N* = 9–15 mice per group). Data are presented as mean ± standard deviation. Abbreviations: *a′* wave mitral annulus tissue motion velocity during atrial contraction; *e′* wave mitral annulus tissue motion velocity during the early ventricular filling phase; *A* wave, mitral valve blood flow velocity during atrial contraction; *E* wave, mitral valve blood flow velocity during the early ventricular filling phase; IVSd, interventricular septum diameter at end diastole; LVIDd, left ventricular internal diameter at end diastole; LVPWd left ventricular posterior wall thickness at end diastole; RWT, relative wall thickness; MV mitral valve.

## Loaded cardiomyocyte sarcomere shortening and lengthening

Immediately after non-loaded cardiomyocyte recordings, glass fibres were attached to cardiomyocyte longitudinally separated positions to permit auxotonic, loaded contraction measurements. For cells successfully attached, the mean difference in sarcomere length measured pre- and post-conversion to loaded mode was <0.02% diastolic length, indicating non-loaded and loaded sarcomere initial lengths were well matched. For attached cardiomyocytes, a comparison of sarcomere shortening behaviour in the loaded state could be made (using FFT) to directly match with data obtained in the non-loaded state.

In the first instance, data from all myocytes were pooled to investigate the general effects of load regardless of treatment group. With loading, for all myocytes (both treatment groups), extent of sarcomere shortening was reduced and shortening and lengthening kinetics slowed when compared with non-loaded performance (Fig. 3*A*). Mean data showed that in response to loading, sarcomere shortening was reduced by ∼34% (Fig. 3*B*, non-loaded: 0.075 ± 0.042, loaded: 0.050 ± 0.031 μm). Maximum rate of shortening was decreased by 58% (non-loaded: −2.85 ± 1.40, loaded: −1.20 ± 0.89 μm/s; not shown), and maximum rate of lengthening by 66% (Fig. 3*C*, non-loaded: 1.46 ± 1.03, loaded: 0.52 ± 0.50 μm/s). The magnitude of the change in extent of shortening and

maximum rate of lengthening with loading were positively correlated (Fig. 3*D*) suggesting the possibility of common underlying mechanisms.

To assess cardiomyocyte rate kinetics in non-loaded and loaded cardiomyocytes independent of shortening extent, the maximum rates of lengthening and shortening were normalized by shortening amplitude to derive 'deformation rate' – that is, rate of length change, relative to extent of shortening ($(\mu m \times s^{-1})/\mu m = 1/s$). The effect of loading on deformation rate was accentuated in the HFSD group (significant ANOVA diet × load interaction effect, Fig. 3*E*). A selective HFSD effect was revealed in the differential between matched 'loaded' and 'non-loaded' lengthening cardiomyocytes. The decrement in deformation rate (lengthening) for CTRL and HFSD is shown in Fig. 3*F*, and is accentuated in HFSD in response to loading (CTRL: 6.84 ± 4.08, HFSD: 12.35 ± 7.15 1/s). Treatment differences in deformation rate were not evident during the shortening phase (data not shown). These findings indicated an accentuated influence of load, specific to the lengthening (i.e. diastolic) phase in HFSD cardiomyocytes and could reflect impaired subcellular relaxation processes specific to the HFSD myocardium. Based on these data identifying different CTRL and HFSD cardiomyocyte responses in the loaded state, we next investigated the effect of stretch applied to cardiomyocytes to simulate *in vivo* 'pre-load' conditions.

### Cardiomyocyte force development: measuring strain and stress during progressive cardiomyocyte stretch

A hallmark of diastolic dysfunction is an elevated diastolic pressure–volume relationship which describes the increased resistance of the myocardial wall to chamber filling (Aurigemma et al., 2006). We sought to design a single cardiomyocyte version of this relationship (Fowler et al., 2015), by applying standardized stretch steps to cardiomyocytes, defining the steps as a percentage of non-stretched state (i.e. 'strain'). Diastolic force was measured simultaneously during length stretch steps. The force measured at every strain increment was normalized by cross sectional area (CSA) to produce a measure of

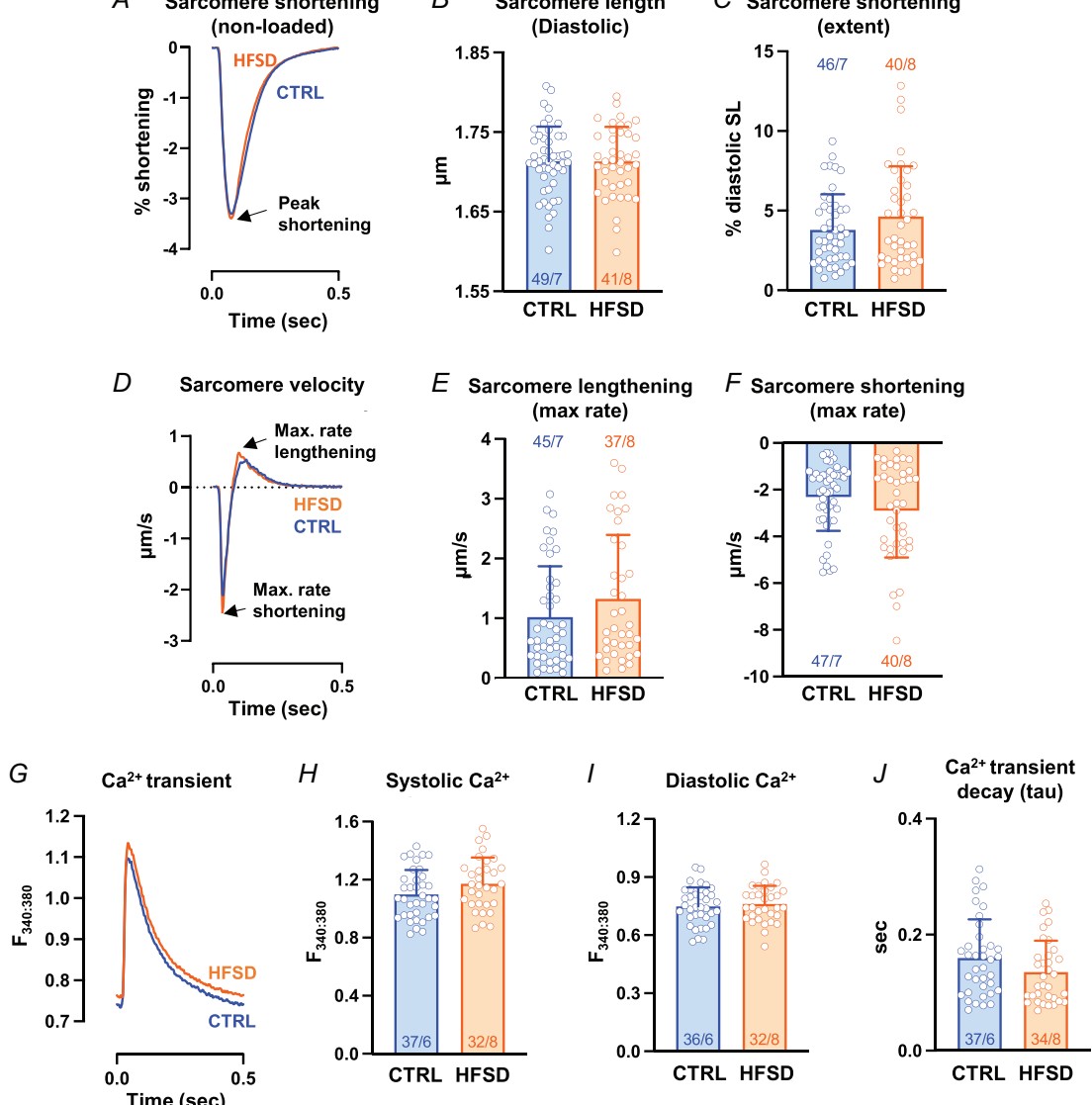

**Figure 2. Non-loaded sarcomere and Ca²⁺ dynamics in CTRL and HFSD cardiomyocytes**
*A*, mean sarcomere contraction cycles presented as a percentage of diastolic sarcomere length for CTRL and HFSD cardiomyocytes. *B* and *C*, mean diastolic sarcomere length and extent of sarcomere shortening for CTRL and HFSD cardiomyocytes. *D*, mean sarcomere shortening and lengthening rate records (time derivative of sarcomere contraction–relaxation cycle) for CTRL and HFSD cardiomyocytes. The nadir and zenith represent maximal shortening and lengthening rates respectively. *E* and *F*, maximal rate of lengthening and shortening rates for CTRL and HFSD cardiomyocytes. *G*, mean Ca²⁺ transient records for CTRL and HFSD cardiomyocytes. *H–J*, mean Ca²⁺ levels (systolic and diastolic) and time constant of transient decay (τ) in HFSD and CTRL cardiomyocytes. Mean data computed from average of 20 cycles per cardiomyocyte to derive group mean (*N* = 6–8 mice per group, *n* = 32–49 cells per group). Comparisons between two treatment groups performed using Student's *t* test (all *P* > 0.05, ns). Data presented as means ± standard deviation. [Colour figure can be viewed at wileyonlinelibrary.com]

'stress' (i.e. force/CSA, mN/mm$^2$). Thus, a sarcomere 'stress/strain' relation (i.e. force-length) was constructed for each myocyte.

The protocols employed to generate stress–strain are depicted in Fig. 4 and described in detail in Methods. Figure 4 shows the stretch standardization approach used to enable reproducible application of strain (converting voltage signals to linear displacement) as a percentage length change (via individual myocyte calibration

constant calculation) to allow comparisons between different cardiomyocytes. The size of each step change in length on sarcomere length was measured (FFT) and confirmed by tracking the intra-cardiomyocyte segment between the glass fibres (Fig. 4). In some cases, a direct sarcomere FFT measure was not microscopically resolvable due to myocyte contraction responses which partially positioned some area of the myocyte outside the optimal focal plane. In these situations, it was

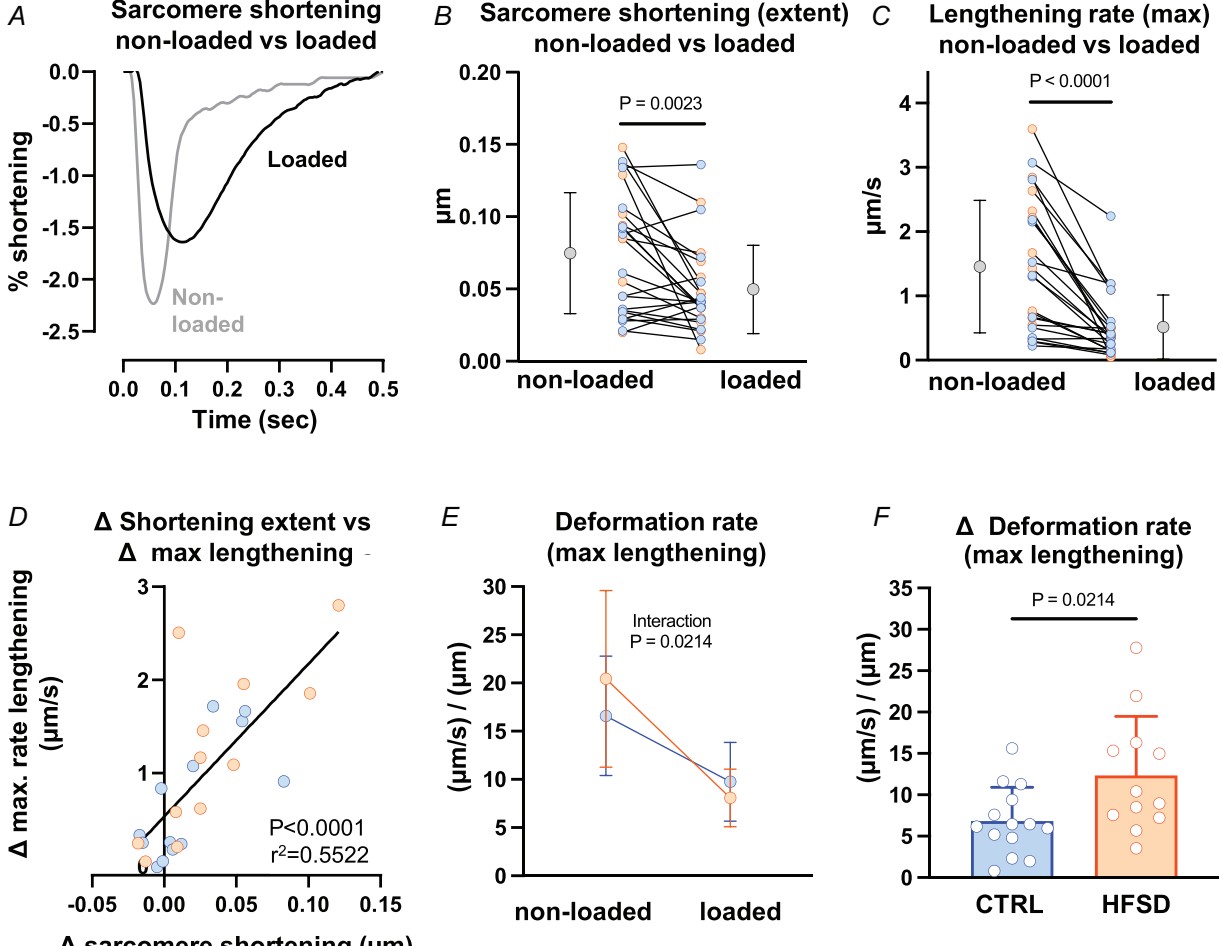

**Figure 3. Sarcomere shortening and lengthening in non-loaded and loaded pacing states**
*A*, exemplar non-loaded and loaded sarcomere contraction cycles for the same cardiomyocyte (CTRL). *B* and *C*, extent of sarcomere shortening and maximum rate of lengthening, measured for each cardiomyocyte in non-loaded and loaded state for CTRL and HFSD groups combined (*N* = 9 mice, *n* = 25 cells). Mean data for CTRL and HFSD are presented as grey points to the left (non-loaded) and right (loaded) of individual cardiomyocyte data. *D*, correlation of loading associated sarcomere shortening change and maximum rate of lengthening change for CTRL and HFSD combined. *E*, mean loading effect on maximum deformation rate during lengthening ((μm/s)/μm; maximum rate of lengthening (μm/s) normalized by shortening extent (μm)) for CTRL *vs*. HFSD (CTRL: *N* = 5 mice, *n* = 14 cells; HFSD: *N* = 4 mice, *n* = 12 cells). Significant ANOVA interaction effect (diet treatment × load state). *F*, mean decrement in deformation rate during lengthening ((μm/s)/μm). Comparison of cardiomyocyte responses in different loading conditions analysed by paired *t* test. Comparison of cardiomyocytes from different treatment groups analysed by unpaired *t* test. A 2-way ANOVA with repeated measures used for analysis of two independent variables. Data fitted with linear regression model and calculation of Pearson's correlation coefficient. Data presented as means ± standard deviation. [Colour figure can be viewed at wileyonlinelibrary.com]

always possible to calculate segment length change. For the example shown, stretch steps were determined to be 2.75 μm, and converted to percentage sarcomere length (i.e. strain) using sarcomere count within intracellular region of interest area (Fig. 4*C*). A comparison

of the variability of cell stretches performed using a uniform (pre-set) stretch constant and the individually derived stretch calibration constant demonstrates that reproducibility is optimized with usage of the individual stretch calibration constant data (Fig. 4*D*). Force,

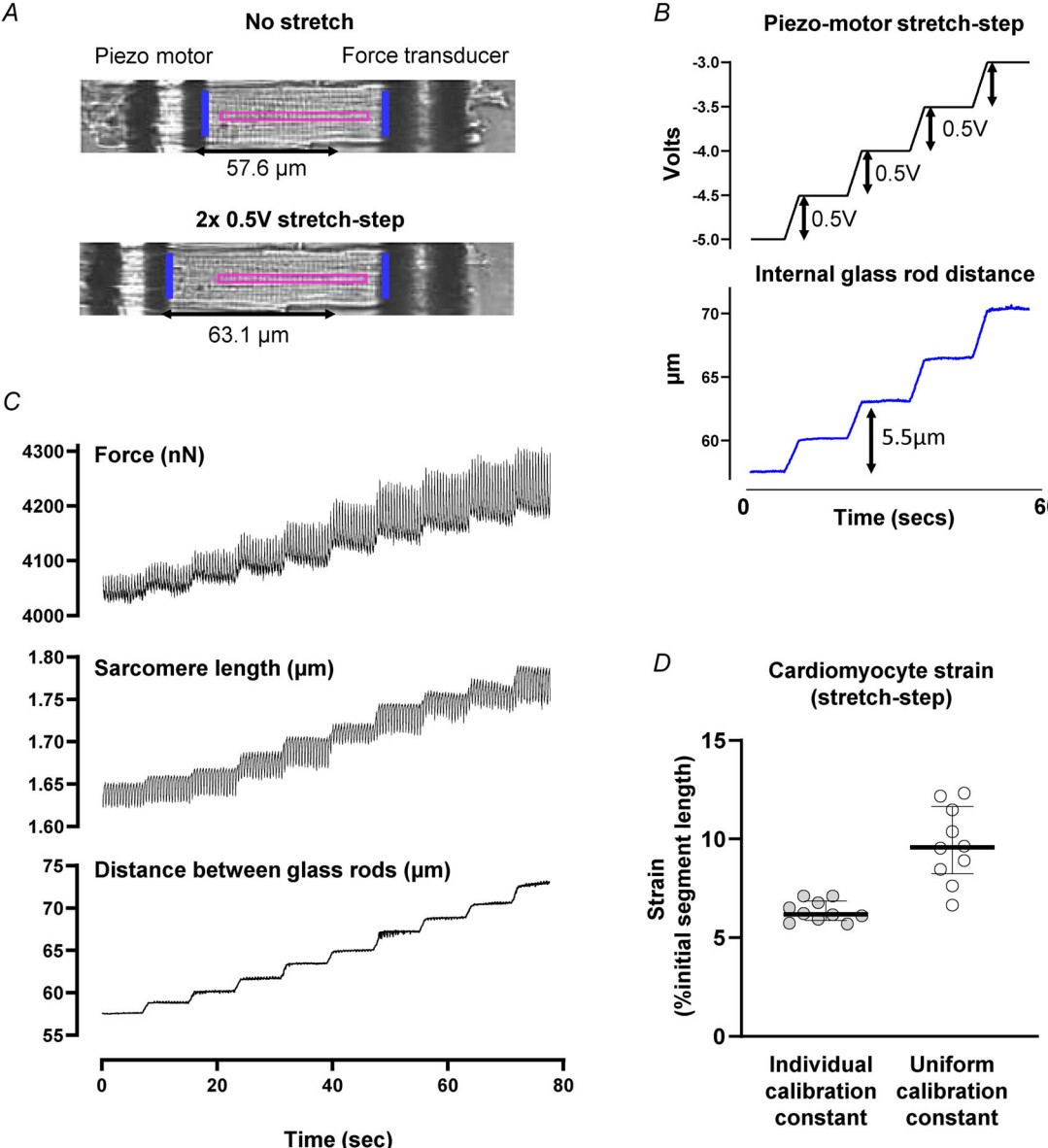

**Figure 4. Cardiomyocyte standardized stretch protocol**
*A*, image of a left ventricular cardiomyocyte longitudinally attached to glass rods. Edge-tracking used to measure inter-rod distance (marked as blue bars) for cardiomyocyte segment analyses. The delineated region of interest (marked pink) represents the area from which fast Fourier transform analysis of sarcomere striations was performed to derive sarcomere-specific data. *B*, exemplar traces depicting the change in glass rod internal distance in response to successive 0.5 V piezo motor inputs used for calibration of stretch protocol for each individual cardiomyocyte to allow normalized comparisons. *C*, measurements of stress and strain during progressive cardiomyocyte stretch. Exemplar force, sarcomere length and rod edge-tracker traces during the first stretch of a stress–length protocol. *D*, comparison of variability of cell stretches performed using a uniform (pre-set) stretch constant and the individually derived stretch calibration constant. Strain was measured as the percentage change in segment length relative to the initial segment length. Reproducibility is optimized using the individual stretch calibration constant. Data shown as medians ± interquartile range. [Colour figure can be viewed at wileyonlinelibrary.com]

sarcomere length and percentage stretch were measured simultaneously (Fig. 4).

### Quantifying diastolic stiffness in CTRL *vs.* HFSD cardiomyocytes using strain normalization methods

Superimposed exemplar stress recordings are shown in Fig. 5*A* for CTRL and HFSD cardiomyocytes, captured during a series of progressive stretches during a 100 s period. The dashed lines are aligned with the diastolic stress step increments for each of the two myocytes (using the data points collected prior to the transition to a new step) throughout the series of progressive stretches. A robust increase in systolic force with stretch is also evident, indicative of length-dependent activation, the cardiomyocyte basis for the Frank–Starling mechanism.

In this experimental setting, stiffness could be visualized as the gradient of the linear plot derived from the cardiomyocyte diastolic stress–length relation. Our stretch protocol methodology provided two approaches for stiffness calculation (as per Methods). A first approach used FFT measurement of sarcomere length to track cardiomyocyte stretch and was plotted according to the standardized stretch steps (in μm). An alternative approach used percentage length change calculated as the length of the cardiomyocyte segment between the glass attachment fibres divided by the initial distance between the fibres prior to stretch (as percentage initial length).

Applying the first approach, a linear regression was performed to fit the data from which a slope value could be derived. Figure 5*B* depicts the average diastolic and systolic stress–sarcomere length relations for CTRL cardiomyocytes only. To determine cardiomyocyte diastolic stiffness using FFT-computed sarcomere length values, the gradient of the mean slope of the diastolic stress sarcomere–length relation for each treatment group was calculated, using mean sarcomere length change (μm) as the denominator. This gradient was significantly higher in HFSD myocytes compared to CTRL (Fig. 5*C*). The individual slope gradients were used to calculate the mean diastolic stress/strain gradient (Fig. 5*D*, CTRL: 2.58 ± 2.25, HFSD: 5.50 ± 4.01 (mN/mm$^2$)/μm, $P = 0.0996$) where there was a trend for a 2.3-fold increase in stiffness of HFSD cardiomyocytes. The statistical power of this analysis was limited due to the difficulty in achieving resolvable diastolic sarcomere lengths by FFT for all myocytes.

Applying the second approach, it was possible to construct stress–segment length (%) relations for all myocytes from which stress data could be obtained. In Fig. 5*E*, the stress–segment length data for HFSD and CTRL are shown, with a robustly significant difference detected in mean gradient values. In Fig. 5*F* the diastolic stress/strain values calculated for the myocytes from the

two treatment groups using this segment approach are directly compared. The HFSD cardiomyocyte stiffness index derived using this approach was increased by 70% compared with CTRL (Fig. 5*F*, CTRL: 0.0075 ± 0.0064, HFSD: 0.0128 ± 0.0068 (mN/mm$^2$)/% length). There was no significant difference in the 'Frank–Starling gain index' (Table 2, CTRL: 2.40 ± 1.06, HFSD: 2.77 ± 1.45, (mN/mm$^2$)/μm) or the ratio of systolic to diastolic stress segment length relations (Table 2, CTRL: 1.62 ± 0.25, HFSD: 1.73 ± 0.50, (mN/mm$^2$)/% length) indicating that cardiomyocyte length-dependent activation is preserved (Bollensdorff et al., 2011; Najafi et al., 2016).

To examine the relationship between intrinsic *in vitro* cardiomyocyte stiffness and *in vivo* diastolic dysfunction, diastolic stiffness (slope of the diastolic stress–segment length relation) averaged per animal was plotted against the $E/e'$ index of diastolic dysfunction. There was a significant correlation between cardiomyocyte stiffness and $E/e'$ in the HFSD group and no evidence of a relationship for the CTRL group (Fig. 5*G* CTRL: $P =$ ns, HFSD: $P = 0.0096$, $r^2 = 0.9213$). These data support the interpretation that intrinsic cardiomyocyte stiffness represents a definable and inherent component of diastolic dysfunction occurring *in vivo* in cardiometabolic disease.

### Load dependence of Ca$^{2+}$ transient decay in CTRL and HFSD cardiomyocytes

To further understand the cardiomyocyte functional elements comprising increased intrinsic stiffness in cardiopathology, the relationship between Ca$^{2+}$ transient features and myofilament deactivation in response to progressive stretch was evaluated. In the intact cardiomyocyte model these electro-mechanical coupling processes are functional for interrogation (in contrast to the disrupted cytosolic environment which characterizes permeabilized myocyte preparations). The Ca$^{2+}$ transient decay features were compared for non-loaded and loaded basal states and also in response to progressive stretch.

In CTRL cardiomyocytes there was no difference in the time constant of Ca$^{2+}$ transient decay ($\tau$, $1/e$) between loaded and stretched state (20% increase in length, Fig. 6*A*; loaded 1.18 ± 0.48; stretched 1.12 ± 0.35-fold change *vs.* non-loaded reference 1.00 ± 0.46). In HFSD cardiomyocytes the time constant of Ca$^{2+}$ transient decay was increased in stretched cardiomyocytes relative to non-loaded (Fig. 6*A*; non-loaded 1.00 ± 0.42; loaded 1.22 ± 0.50; stretched 1.42 ± 0.65 fold change). Loading prolonged time to peak Ca$^{2+}$ transient in both CTRL and HFSD cardiomyocytes (Fig. 6*B*; CTRL: non-loaded 1.00 ± 0.25; loaded 1.49 ± 0.38; stretched 1.46 ± 0.44, HFSD: non-loaded 1.00 ± 0.41; loaded 1.36 ± 0.41; stretched 1.21 ± 0.34-fold change). The effects can be

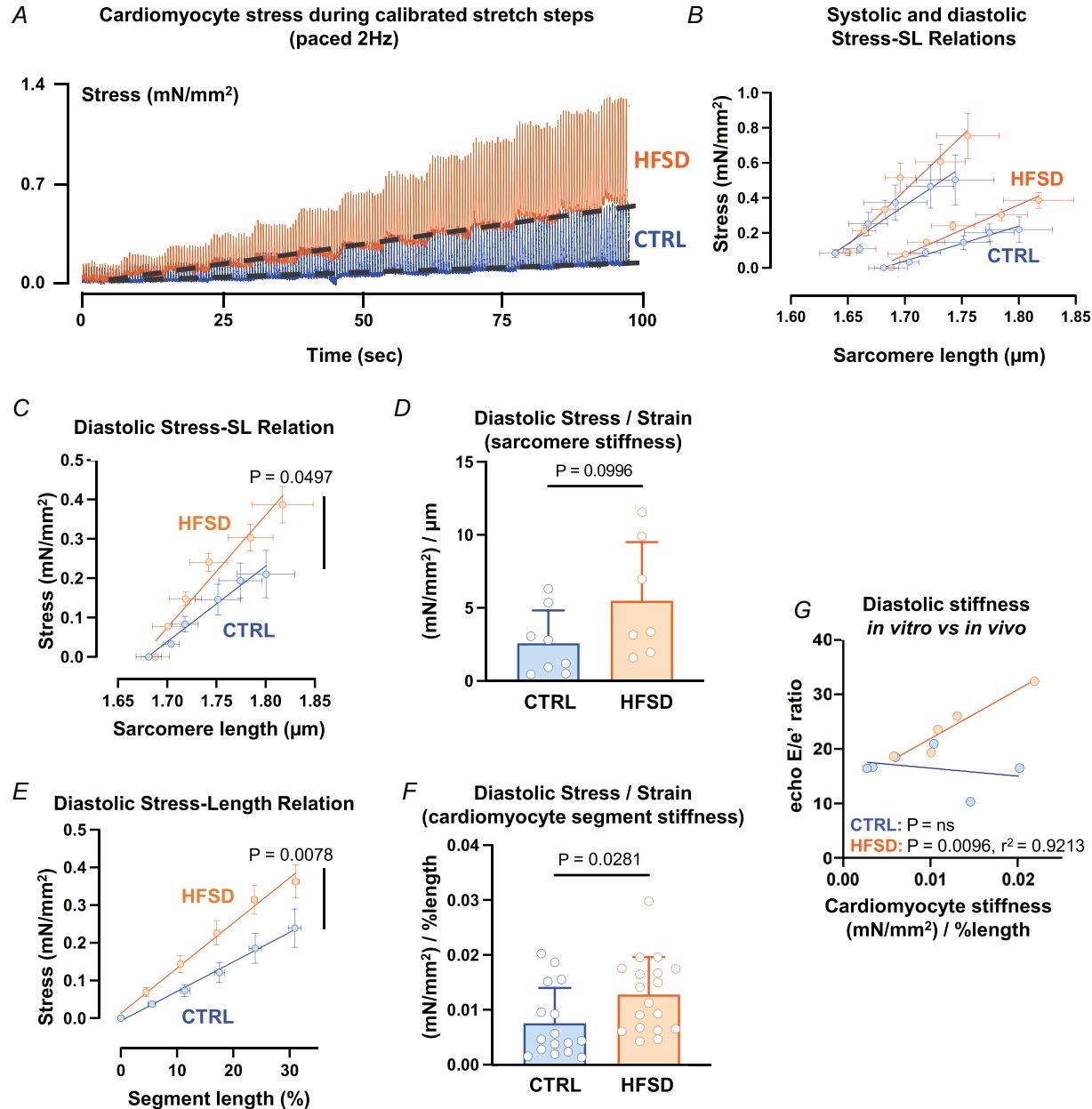

**Figure 5. Quantification of diastolic stiffness in CTRL *vs.* HFSD cardiomyocytes using strain normalization methods**

*A*, exemplar stress recordings of CTRL and HFSD cardiomyocytes undergoing 'stress–length' protocol (2 Hz paced). Diastolic baseline stress shifts indicated by dashed line. *B*, diastolic (pair of lower lines) and systolic (pair of upper lines) sarcomere mean stress–sarcomere length (SL) relations in CTRL cardiomyocytes ($N$ = 5 mice, $n$ = 9 cells). *C*, diastolic mean stress–sarcomere length relation (linear regression fit) for HFSD and CTRL. *D*, diastolic stress (mN/mm$^2$) normalized relative to sarcomere stretch (µm) yields sarcomere stress/strain = stiffness ((mN/mm$^2$)/strain (µm)) for HFSD and CTRL. *E*, diastolic mean cardiomyocyte segment stress–length relation (linear regression fit) for HFSD and CTRL. *F*, diastolic stress (mN/mm$^2$) normalized relative to segment stretch (% length) yields cardiomyocyte segment stress/strain = stiffness ((mN/mm$^2$)/strain (% length)) for HFSD and CTRL. *G*, correlation (linear regression fit) of *in vitro* cardiomyocyte segment diastolic stiffness (stress/strain) with *in vivo* diastolic dysfunction *E/e*′ (blood flow/wall motion) in CTRL and HFSD. Cardiomyocyte data averaged per heart. Comparisons between two groups performed using Student's *t* test. Data fitted with linear regression model and calculation of Pearson's correlation coefficient, presented as means ± standard deviation (*D*, *F*) and means ± standard error of the mean (*B*, *C*, *E*). For *C* and *D*, CTRL: $N$ = 4 mice, $n$ = 8 cells; HFSD: $N$ = 2 mice, $n$ = 7 cells. For *E* and *F*, CTRL: $N$ = 6 mice, $n$ = 16 cells; HFSD: $N$ = 6 mice, $n$ = 18 cells. [Colour figure can be viewed at wileyonlinelibrary.com]

**Table 2. Stress–length relation mean values for CTRL and HFSD mouse cardiomyocytes**

| | CTRL 2 Hz | HFSD 2 Hz | *P* |
|---|---|---|---|
| Systolic stress–SL relation slope (($mN/mm^2$)/$\mu m$) | $6.54 \pm 4.97$ | $16.35 \pm 15.09$ | 0.087 |
| Systolic stress–length relation slope (($mN/mm^2$)/%length) | $0.013 \pm 0.011$ | $0.022 \pm 0.011$ | 0.022 |
| Systolic: diastolic stress–SL relation 'Frank–Starling gain index' (ratio) | $2.40 \pm 1.06$ | $2.77 \pm 1.45$ | 0.562 |
| Systolic: diastolic stress–length relation (ratio) | $1.62 \pm 0.25$ | $1.73 \pm 0.50$ | 0.471 |

Comparisons between two treatment groups performed using Student's unpaired *t* test (CTRL: *N* = 4–6 mice, *n* = 9–17 cells; HFSD: *N* = 2–6 mice, *n* = 7–19 cells). Data are presented as means ± standard deviation.

visualized in Fig. 6*C* and *D* where the mono-exponential decay of exemplar $Ca^{2+}$ transients is depicted. The shallower decay of the $Ca^{2+}$ transient in the stretched state (Fig. 6*D*, HFSD) was apparent whereas the loaded and non-loaded decay traces were virtually overlaid for CTRL cardiomyocytes (Fig. 6*C*). These data suggest that stretch may enhance a mechanism by which the myofilaments bind $Ca^{2+}$ for a prolonged duration during relaxation thus reducing $Ca^{2+}$ availability for extrusion from the cytosol.

## Pacing-induced $[Ca^{2+}]_i$ increase to examine potential for a cross-bridge derived stiffness mechanism

A further mechanistic study to probe the potential relationship between $Ca^{2+}$ and stiffness involved perturbation of cytosolic $Ca^{2+}$ cycling time. Elevating stimulation frequency is a recognized physiological strategy to increase diastolic $Ca^{2+}$ levels and to apply increased cardiomyocyte workload. In response to frequency elevation (2–4 Hz), cardiomyocyte diastolic $Ca^{2+}$ levels increased in both CTRL and HFSD

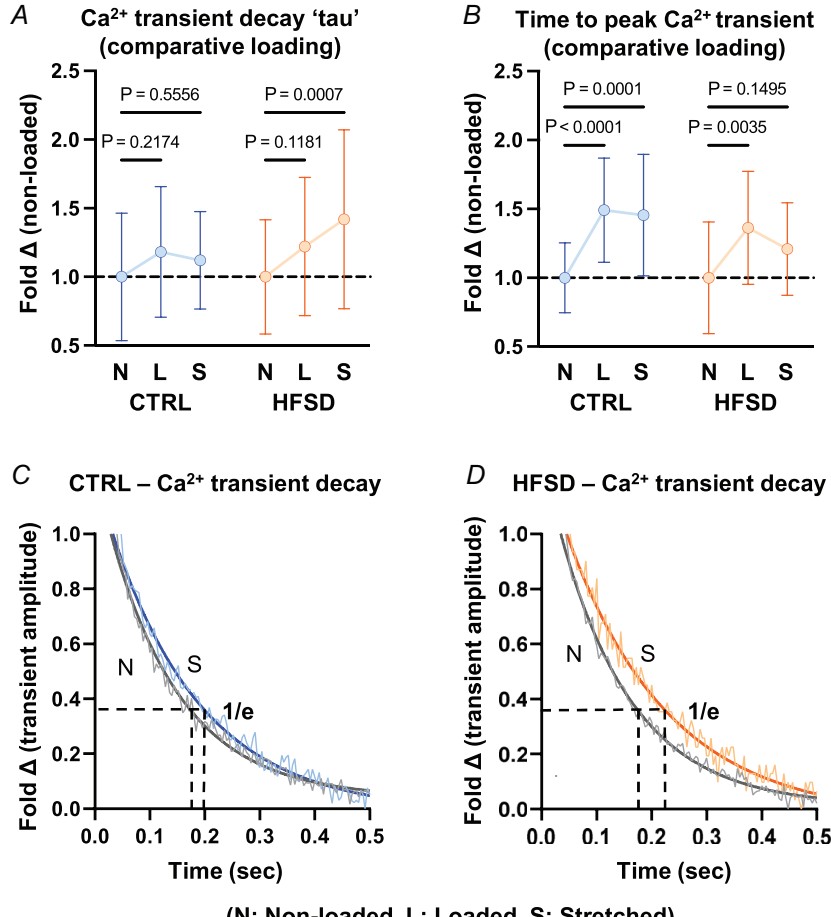

**(N: Non-loaded, L: Loaded, S: Stretched)**

**Figure 6. Load dependence of cardiomyocyte $Ca^{2+}$ transient decay in CTRL and HFSD cardiomyocytes**

*A* and *B*, time constant of $Ca^{2+}$ transient decay ($\tau$) and time to peak $Ca^{2+}$ transient in cardiomyocytes with different load conditions: loaded (L) and stretched (S) shown as fold change relative to non-loaded (N) for CTRL and HFSD. *C* and *D*, exemplar $Ca^{2+}$ transient decay records, with a mono-exponential decay function fit, in non-loaded and stretched CTRL and HFSD cardiomyocytes. Horizontal black broken lines intersect with mono-exponential fit at 36.8% (1/e) transient amplitude. Vertical broken lines from intersection points indicate the time constant of $Ca^{2+}$ transient decay ($\tau$). For analysis of two independent variables a 2-way ANOVA with repeated measures used (CTRL: *N* = 5 mice, *n* = 13 cells; HFSD: *N* = 4 mice, *n* = 12 cells). Data are presented as means ± standard deviation. [Colour figure can be viewed at wileyonlinelibrary.com]

(Fig. 7*A* and *B*) groups consistent with previous studies (Eisner et al., 2020; Varian & Janssen, 2007). In the HFSD cardiomyocytes this effect was markedly diminished and about half the mean magnitude of the $Ca^{2+}$ shift observed for CTRL myocytes (Fig. 7*B*, CTRL 2 Hz: 0.664 ± 0.052, CTRL 4 Hz: 0.775 ± 0.083, HFSD 2 Hz: 0.688 ± 0.099, HFSD 4 Hz: 0.741 ± 0.126, $F_{340:380}$, 2-way ANOVA significant interaction effect; Fig. 7*C*, CTRL 0.110 ± 0.051; HFSD 0.054 ± 0.049, $F_{340:380}$). No treatment or interaction effects were observed for systolic $Ca^{2+}$, $Ca^{2+}$ amplitude, or time constant of $Ca^{2+}$ transient decay (Table 3). Frequency elevation was associated with a reduction in diastolic sarcomere length in both CTRL

(3%) and HFSD (5%) loaded cardiomyocytes (Fig. 7*D*; CTRL 2 Hz: 1.69 ± 0.04, CTRL 4 Hz: 1.64 ± 0.05, HFSD 2 Hz: 1.69 ± 0.04, HFSD 4 Hz: 1.61 ± 0.07 μm). The combination of normal diastolic sarcomere length reduction, concomitant with suppressed diastolic $Ca^{2+}$ increase in HFSD cardiomyocytes, may indicate elevated diastolic responsiveness to $Ca^{2+}$ in the HFSD cardiomyocyte.

While frequency elevation did not change the stress/strain relation in CTRL cardiomyocytes, a significant increase in the stress/strain gradient was observed with increased frequency in HFSD cardiomyocytes (Fig. 7*E* and *F*; CTRL 2 Hz: 0.0060 ± 0.0056,

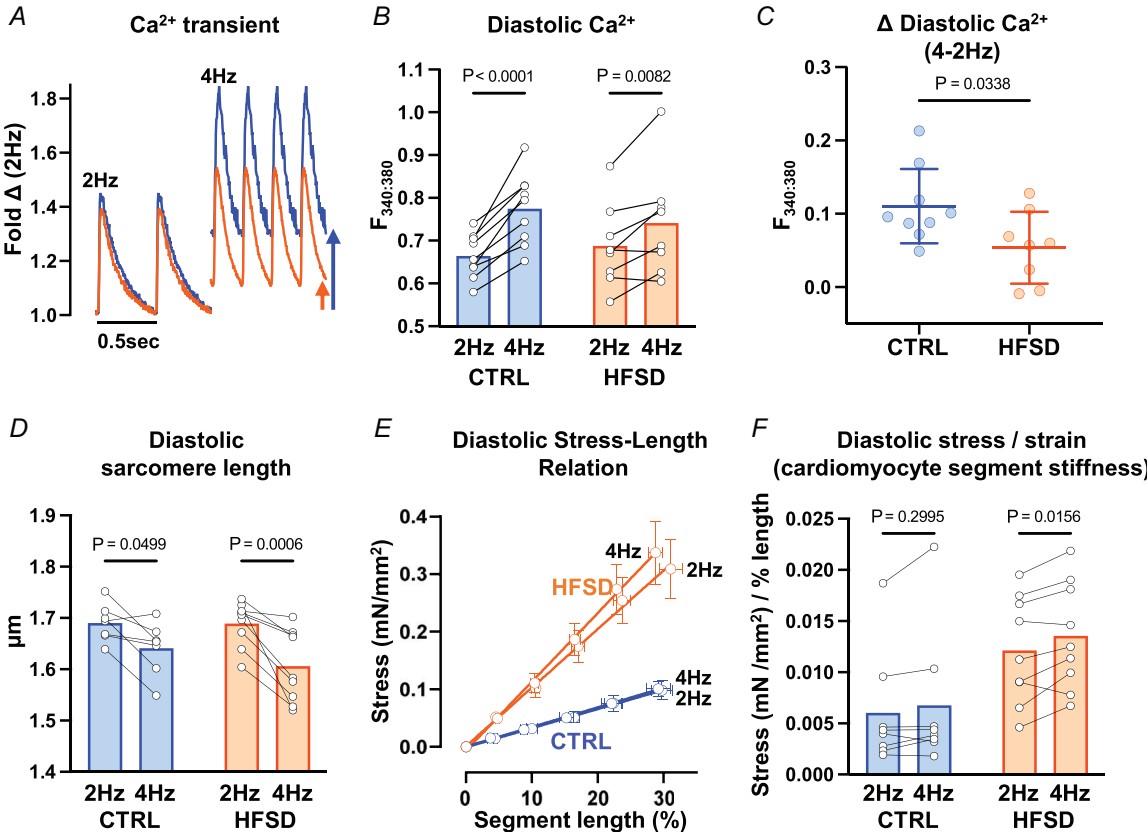

**Figure 7. Pacing induced changes in cardiomyocyte cytosolic $Ca^{2+}$ levels and stiffness in CTRL and HFSD groups**

*A*, exemplar CTRL and HFSD cardiomyocyte $Ca^{2+}$ transients at 2 and 4 Hz pacing. *B*, diastolic cardiomyocyte $Ca^{2+}$ levels measured before and after transition from 2 to 4 Hz pacing in HFSD and CTRL. *C*, diastolic cardiomyocyte $Ca^{2+}$ level mean shift with transition from 2 to 4 Hz pacing in HFSD and CTRL. *D*, diastolic sarcomere length measured before and after transition from 2 to 4 Hz pacing in HFSD and CTRL. *E*, diastolic mean cardiomyocyte segment stress–length relation (linear regression fit) for HFSD and CTRL at 2 and 4 Hz pacing. *F*, diastolic mean cardiomyocyte segment stress–length relation slope (mN/mm²) normalized relative to segment stretch (% length) yields cardiomyocyte mean segment stress/strain = stiffness ((mN/mm²)/strain (% length)) for HFSD and CTRL at 2 and 4 Hz pacing. Comparison between two groups performed using unpaired *t* test. With two independent variables a repeated measures 2-way ANOVA Šidák's *post hoc* analysis was performed. Data are presented as means ± standard deviation (*C*), means ± standard error of the mean (*E*) or mean with paired individual points (*B*, *D*, *F*). For *B* and *C*, CTRL: *N* = 4 mice, *n* = 9 cells; HFSD: *N* = 4 mice, *n* = 8 cells. For *D* and *E*, CTRL: *N* = 4 mice, *n* = 7 cells; HFSD: *N* = 4 mice, *n* = 9 cells. For *F*, CTRL: *N* = 4 mice, *n* = 8 cells; HFSD: *N* = 4 mice, *n* = 9 cells. [Colour figure can be viewed at wileyonlinelibrary.com]

**Table 3. Ca$^{2+}$ transient parameters for loaded CTRL and HFSD mouse cardiomyocytes**

| | CTRL | | HFSD | | P | | |
|---|---|---|---|---|---|---|---|
| | 2 Hz | 4 Hz | 2 Hz | 4 Hz | Frequency | Diet | Interaction |
| Systolic Ca$^{2+}$ ($F_{340:380}$) | 0.93 ± 0.08 | 1.05 ± 0.10# | 0.99 ± 0.15 | 1.08 ± 0.20 | 0.0004 | 0.571 | 0.100 |
| Amplitude Ca$^{2+}$ ($F_{340:380}$) | 0.26 ± 0.06 | 0.28 ± 0.06 | 0.32 ± 0.10 | 0.33 ± 0.11 | 0.569 | 0.0672 | 0.723 |
| Time constant of Ca$^{2+}$ decay ($\tau$, s) | 0.21 ± 0.07 | 0.09 ± 0.02# | 0.17 ± 0.07 | 0.09 ± 0.03# | <0.0001 | 0.207 | 0.350 |

Comparisons in data with two independent variable were performed using a mixed analysis 2-way ANOVA with Šidák's *post hoc* analysis (CTRL, 2 Hz: *N* = 5 mice, *n* = 13 cells; CTRL, 4 Hz: *N* = 4 mice, *n* = 9 cells; HFSD, 2 Hz: *N* = 5 mice, *n* = 14 cells; HFSD, 4 Hz: *N* = 4 mice, *n* = 8 cells). Data are presented as means ± standard deviation. *P*-values for ANOVA effects (Frequency, Diet, Interaction) are listed within the table. #Significant pairwise *post hoc* frequency differences *vs.* CTRL 2 Hz or HFSD 2 Hz, *P* < 0.05. No pairwise *post hoc* treatment differences were observed.

CTRL 4 Hz: 0.0068 ± 0.0068; HFSD 2 Hz: 0.0121 ± 0.052, HFSD 4 Hz: 0.0136 ± 0.0052 (mN/mm$^2$)/% length). This may indicate a larger population of cross-bridges contributing to diastolic stiffness at heightened workloads in HFSD cardiomyocytes although other potential non-cross-bridge sources cannot be excluded. Since Ca$^{2+}$ shift in response to increased pacing is modest (relative to CTRL), increase in stiffness cannot be attributed to rise in Ca$^{2+}$ levels *per se*. Shift in the myofilament–Ca$^{2+}$ response during diastole is a potential alternative explanation.

## Discussion

Here we provide the first quantitative analysis of intact cardiomyocyte stiffness and performance deficits associated with acquired cardiometabolic disease. We demonstrate that a component of cardiac diastolic dysfunction in cardiometabolic disease derives from intrinsic cardiomyocyte mechanical abnormality.

The concept of myocardial stiffness is a frequently invoked qualitative descriptor of the pathophysiological phenotype characterizing diastolic dysfunction. Clinically and experimentally the echocardiographic parameter *E/e′* is used as an index of myocardial stiffness in disease diagnosis. *E/e′* is a compound measure of cellular and extracellular matrix stiffness contributors. Despite consistent *in vivo* findings, an understanding of the cardiomyocyte-specific contribution to tissue stiffness has been difficult to identify. Using a series of intact single cell recording manoeuvres, we have established that the stiffness characteristics of individual isolated cardiomyocytes from normal and dysfunctional hearts can be defined, normalized and compared. Further, a significant relationship was identified between the mean stiffness of single cardiomyocytes and the extent of diastolic dysfunction exhibited by the hearts from which they were derived (i.e. *E/e′ vs.* Young's modulus; (mN/mm$^2$)/% length).

To recapitulate, cardiometabolic disease induced by dietary intervention in mice was characterized by marked body weight elevation (50%, 'obesity') and glucose intolerance. *In vivo* echocardiography diagnosed significant diastolic dysfunction with a 35% elevation of the *E/e′* in HFSD animals in the absence of systolic functional deficit. In paced non-loaded conditions, the sarcomere contraction cycles and Ca$^{2+}$ transients (fura-2) were similar in CTRL and HFSD cardiomyocytes. When the same cardiomyocytes were loaded, the shortening extent and maximal lengthening rate were reduced for both diet conditions. In HFSD cardiomyocytes the reduction in deformation rate during lengthening (but not shortening) was accentuated, indicating a selective negative impact of loading on relaxation of the HFSD-derived cardiomyocytes. In HFSD cardio-

myocytes there was a larger increase in diastolic force per cross-sectional area (stress: $mN/\mu m^2$) in response to standardized step increases in length (strain: % initial length ($\mu m$)). From these data a 'stress/strain' parameter was determined, defining diastolic stiffness for each myocyte i.e. 'Youngs modulus' (($mN/\mu m^2$)/% initial length ($\mu m$)). Mean stiffness was 70% higher in cardiomyocytes of HFSD animals. Thus, the increase in cardiomyocyte stiffness index was approximately double the elevation of the *in vivo* $E/e'$ dysfunction index. Using matched *in vivo* and *in vitro* data, a significant correlation was found between diastolic dysfunction ($E/e'$) and mean cardiomyocyte stiffness value for HFSD animals, while there was no relationship between these parameters in the myocytes of the CTRL-diet fed group. To evaluate the possible influence of mechanical state on the $Ca^{2+}$ transient, the mean transient decline parameters ($\tau$) from non-loaded, loaded and stretched cardiomyocytes were compared. In HFSD cardiomyocytes the time constant of $Ca^{2+}$ transient decay was increased in stretched cardiomyocytes relative to non-loaded, a feature not observed in CTRL cardiomyocytes. Finally, in response to an *in vitro* pacing increase, HFSD cardiomyocytes exhibited accentuated diastolic stiffness even with a smaller increase in diastolic $Ca^{2+}$ level. Collectively, these findings suggest a previously undescribed alteration in $Ca^{2+}$–myofilament interactions indicative of a cross-bridge component to cardiomyocyte stiffness in cardiometabolic disease.

### Dietary model of acquired metabolic syndrome and cardiac diastolic disease

High fat/sugar diet feeding (i.e. 'western diet') has previously been shown experimentally to induce diastolic dysfunction in rodents. Elevated $E/e'$ is a remarkably consistent echocardiographic feature of high fat/sugar diet-fed mouse myocardial performance while E/A and IVRT may be elevated, reduced or unchanged relative to control counterparts (Daniels et al., 2022; Dia et al., 2020; Pulinilkunnil et al., 2014; Sowers et al., 2020; Wingard et al., 2021). The dietary regime employed in this study has been well characterized as a model of diastolic dysfunction, with gradual development of metabolic disturbance (stable rate of weight gain and mild hyperglycaemia), with maintained systolic function (ejection fraction and fractional shortening). The data presented here are consistent with clinical observations linking cardiometabolic disease and diastolic dysfunction.

As we have reported previously, over the time span of the dietary intervention, differential fibrosis is not observed (Daniels et al., 2022; Wells et al., 2023). The finding of increased stiffness in cardiometabolic disease is consistent with some aspects of prior experiments involving animal feeding of obesogenic diets, using other

rodent preparation types – including permeabilized cardiomyocytes, tissue strips/papillary muscles, and non-paced cardiomyocytes (Gonçalves et al., 2016; Leopoldo et al., 2010; Sowers et al., 2020). Interestingly stiffness in those studies was attributed to titin or extracellular matrix properties. Our protocol with intact single cardiomyocytes allows a mechanistic extension of these previous findings.

### Loading the intact cardiomyocyte – functional deficits revealed in HFSD mice

Non-loaded contracting cardiomyocytes of CTRL and HFSD mice exhibited identical shortening–lengthening cycles, but a distinctly different decrement in lengthening rate when exposed to load (i.e. auxotonic with progressive 'pre-load' increase). Our findings show unequivocally that cardiomyocyte mechanical dysfunction cannot be detected by analysis of non-loaded shortening. With loading, the shortening extent and the maximal lengthening rates were reduced for both CTRL and HFSD. In HFSD the lengthening rate reduction was accentuated even when shortening extent was accounted for. These findings indicated specific influence of loading on lengthening in HFSD cardiomyocytes and could reflect impaired subcellular relaxation processes specific to the HFSD myocardium.

Others have demonstrated a physiological effect of loading on shortening in cardiac cells/tissues in different species (using slow pacing and sub-physiological temperature conditions), although the pronounced effect on relaxation is not consistently reported (Lab et al., 1984; Shimkunas et al., 2021; White et al., 1995). This is consistent with *in vivo* human clinical observations where there is a strong inverse association between late-systolic loading and left ventricular wall displacement velocity during early diastole (Borlaug et al., 2007). Importantly, increased dependence of relaxation on after-load is reported to be an *in vivo* feature of heart failure with diastolic dysfunction (Eichhorn et al., 1992; Gillebert et al., 2000). Regarding the $Ca^{2+}$ transient, in the current study a subtle prolongation effect is observed in both CTRL and HFSD cardiomyocytes when loaded (but not stretched) which is consistent with recently reported findings in rat LV trabeculae (Dowrick et al., 2022). The variability in the sarcomere shortening recorded for both non-loaded and loaded cardiomyocytes is notable but consistent with other studies assessing relaxation of rodent cardiomyocytes (Clark & Campbell, 2019; Delbridge & Roos, 1997). A range of non-loaded cardiomyocyte mean diastolic sarcomere lengths are reported in the literature. When cells are evaluated *in vitro* and not constrained by local structural elements, sarcomere lengths <1.8 $\mu m$ are commonly observed and consistent

with our findings (Bub et al., 2010; Lim et al., 2000). *In vitro* experimental conditions can also contribute to variation in diastolic sarcomere lengths (e.g. pacing frequency, temperature and buffer tonicity) (Chung & Campbell, 2013; Khokhlova et al., 2022; Roos et al., 1982).

### Characterizing cardiomyocyte stretch & pacing challenge responses

In order to quantify intact single cardiomyocyte stiffness our approach was to apply a calibrated stretch protocol, based on pre-measurement of a stretch calibration constant for each myocyte, tracking changes in force generation and cytosolic $Ca^{2+}$ responses for each stretch step. This calibrated step-by-step approach enabled the stress and strain data to be gathered from actively paced cardiomyocytes over extended periods (i.e. 80–100 s) and computed to generate a definitive, comparative stiffness parameter for each cardiomyocyte.

In the intact myocyte model, the internal cytosolic regulatory milieu is preserved, and the cytoskeletal/sarcomeric/organelle structural framework is integral. Thus, contributions to diastolic 'stiffness' reflect components of the structural passive resistance to deformation and the basal crossbridge-derived activity. Recent studies of permeabilized cardiac muscle strip bundles have explored the separate contributions of mictrotubule, titin, actin-associated, sarcolemmal and desmin-linked components to passive force development (Loescher et al., 2023). Prior studies of 'skinned' or quiescent myocardial preparations have also identified sources of passive stiffness elevation, with titin isoform switch and modification identified as major modulators of stiffness in cardiometabolic disease (Abdellatif et al., 2021; Gonçalves et al., 2016; Hamdani et al., 2013; Hopf et al., 2018; Leopoldo et al., 2010; Sowers et al., 2020).

Our findings indicate involvement of a cytosolic $Ca^{2+}$ component of the stretch response. $Ca^{2+}$ fluorescence analyses from these stretch protocols showed novel stretch-dependent prolongation of $Ca^{2+}$ transient decay in the HFSD cardiomyocytes. The time constant of $Ca^{2+}$ transient decay increased with loading and further with stretching – a feature not evident in CTRL cardiomyocytes and which provides additional mechanistic insight.

The final pacing frequency protocol was implemented as a relative workload stress to the loaded cardiomyocyte. With a doubling of pacing frequency, a significant increase in the stress/strain gradient was observed in HFSD cardiomyocytes (and not CTRL), and a similar reduction in diastolic sarcomere length in both CTRL and HFSD was observed. While cardiomyocyte diastolic $Ca^{2+}$ level increased in both CTRL and HFSD, the HFSD increase was less marked than CTRL. This was a surprising finding, not previously reported in any other model setting. A

suggestion of $Ca^{2+}$ hyper-sensitivity arises (Davis et al., 2007) amongst other explanations to consider.

### Mechanistic insights into cross-bridge derived stiffness

Here we report that a cardiometabolic disease phenotype induced by an obesogenic diet includes a 70% increase in cardiomyocyte stiffness which contributes to a 35% increase in diastolic dysfunction *in vivo* (by echo $E/e'$). Several lines of evidence emerging from this study suggest that $Ca^{2+}$-related mechanism(s) may contribute to cardiomyocyte stiffness increase. Specifically, in the loaded HFSD cardiomyocyte (*vs.* CTRL):

(1) Load confers accentuated lengthening rate deficit.
(2) With stretch, the $Ca^{2+}$ transient decay time course is prolonged.
(3) With more rapid pacing, relatively small increases in diastolic $Ca^{2+}$ are associated with elevated stiffness.

Together these cardiomyocyte phenotypes would be consistent with selective impairment of relaxation rate, selective myofilament hyper-activation, impeded diastolic filling and reduced diastolic reserve.

This combination of observations is consistent with the notion that stretch-dependent cross-bridge attachment is prolonged in HFSD cardiomyocytes (or conversely detachment is delayed). We can speculate that an alteration in the three-dimensional geometry of the low affinity $Ca^{2+}$ binding pocket in troponin C (TnC) would produce this effect, by stabilizing $Ca^{2+}$ in the pocket for longer every activation cycle. This could increase TnC buffering power, which would explain the relatively small $Ca^{2+}$ rise in HFSD in response to increased pacing (Eisner et al., 2023). A manifestation of this altered binding function could be apparent $Ca^{2+}$ hypersensitivity. The TnC–$Ca^{2+}$ dissociation rate has been theorized as an effector of the transient decay time course (Robinson et al., 2018). A role for the troponin complex in regulation of $Ca^{2+}$ affinity is well established (Chemla et al., 2000). Given the load and stretch-dependent nature of these findings, a role for stress sensor proteins such as titin is likely. Indeed, there is some evidence to suggest that titin isoform regulates TnC low-affinity binding pocket opening and $Ca^{2+}$ affinity in response to stretch by regulating the number of attached cross-bridges (Li et al., 2019). Further evidence of structural modifications of specific myofilament regions which form and regulate this pocket is required. Mass spectrometry studies by us and others provide proof of concept of relevant post-translational modification types (Janssens et al., 2018; Loescher et al., 2022).

## Limitations

Given the diastolic dysfunction focus of this study, investigations have primarily examined cardiomyocyte relaxation and lengthening behaviours. Diastolic and systolic interactions are important and additional protocols are in development for these studies. In this regard, we defer to the extensive body of technological research undertaken to develop software and instrumentation to perform single cardiomyocyte 'pressure–volume (PV) loops' (Helmes et al., 2016). Using this approach, to mimic PV cycles in the intact myocardium, stretch is applied after relaxation has occurred by using a force clamp to maintain a predefined diastolic force level. This approach has been used to discover that titin truncation shifts the working sarcomere length range preserving diastolic stiffness at the expense of systolic performance (Methawasin et al., 2022) and that highly compliant titin allows cardiomyocytes to function at high diastolic $Ca^{2+}$ levels (Najafi et al., 2019). Potentially, this approach could also be used to evaluate the temporal relationship between slow or incomplete relaxation and cardiomyocyte cross-bridge derived stiffness in a *semi situ* manner. Operator expertise is imperative with this system, which to date has been very productive in the hands of the developers.

In our work we have adopted a pragmatic approach, simpler to implement – although still very challenging. Our protocol involves the application of cumulative ramp steps with about 20% non-loaded-to-loaded myocyte conversion success rate. When animals have many months of dietary treatment history, and as cell isolation from older and diseased hearts is difficult, an optimized loading and stretching approach is key. The two approaches have not been benchmarked, although it can be predicted that the viscous components of stiffness are likely underestimated when stretches are slow and relatively small such as in the current study (Caporizzo & Prosser, 2021). The capacity to directly and reproducibly determine Young's modulus (stress/strain) with a slow stretch routine has useful comparative value. An additional aspect of our pragmatic approach has been to evaluate cardiomyocyte segment stretches as average sarcomere stretch methods can be challenging to track optically.

With ongoing research, a priority is to undertake longitudinal studies, including both sexes, to investigate the relative timing of the emergence of *in vivo* and *in vitro* stiffness phenotypes as differences in $Ca^{2+}$ handling have previously been observed between control male and female rodents (Farrell et al., 2010). This has high translational relevance as we have previously identified a differential time course of diastolic dysfunction emergence in females and males, with increased female vulnerability (Chandramouli et al., 2018; Reichelt et al., 2013).

## Conclusion

This is the first study to link elevated intact single cardiomyocyte stiffness with *in vivo* diastolic dysfunction in a dietary acquired cardiometabolic disease mouse model. The novel findings identify augmented load dependent TnC–$Ca^{2+}$ interactions as one potential feature of myofilament function which contributes to cardiomyocyte stiffness in hearts exhibiting cardiometabolic diastolic dysfunction. Taken together, the findings emphasize the importance of mechanical loading and pacing frequency in revealing latent diastolic abnormalities at the cardiomyocyte level. The rationale for further studies identifying specific subcellular and molecular mechanisms of cardiomyocyte stiffness in cardiometabolic disease prior to failure onset is compelling.

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

## Additional information

### Data availability statement

The data that support the findings of this study are available from the corresponding author upon reasonable request.

### Competing interests

The authors declare that they have no competing interests.

### Author contributions

All experiments were performed at the Cardiac Phenomics Laboratory at the University of Melbourne, Australia. J.V.J., K.M.M. and L.M.D.D. contributed to the conception and research design of the work. All authors contributed to the acquisition, analysis or interpretation of data for the work and contributed to drafting the work or revising it critically for important intellectual content. All authors have read and approved the final version of this manuscript and agree to be accountable for all aspects of the work in ensuring that questions related to the accuracy or integrity of any part of the work are appropriately investigated and resolved. All persons designated as authors qualify for authorship, and all those who qualify for authorship are listed.

### Funding

This work was supported by grants from the National Health and Medical Research Council of Australia (LMDD: NHMRCA 1157320), the Diabetes Australia Research Trust (LMDD), the National Heart Foundation of Australia (LMDD: NHFA), the New Zealand Marsden Fund (KMM: 19-UOA-268), the Health Research Council of New Zealand (KMM:19/190), and the National Institutes of Health, USA (JVE: R01 HL155346-01 and R01 HL144509-01).

### Keywords

$Ca^{2+}$ myofilament interaction, cardiometabolic disease, cardiomyocyte, diastolic dysfunction, relaxation, stiffness

### Supporting information

Additional supporting information can be found online in the Supporting Information section at the end of the HTML view of the article. Supporting information files available:

**Peer Review History**

