## [Peer Review History · The Journal of Physiology]

Mechanical loading reveals an intrinsic cardiomyocyte stiffness contribution to diastolic dysfunction in murine cardiometabolic disease

Johannes V Janssens, Antonia J.A. Raaijmakers, Parisa Koutsifeli, Kate L Weeks, James R Bell, Jennifer E Van Eyk, Claire L Curl, Kimberley M Mellor, and Lea M. D. Delbridge

DOI: 10.1113/JP286437

Corresponding author(s): Lea Delbridge (imd@unimelb.edu.au)

The following individual(s) involved in review of this submission have agreed to reveal their identity: Rémi Peyronnet (Referee #1)

Review Timeline:

Submission Date:	18-Apr-2024
Editorial Decision:	12-Jun-2024
Revision Received:	29-Sep-2024
Editorial Decision:	31-Oct-2024
Revision Received:	01-Nov-2024
Accepted:	04-Nov-2024

Senior Editor: Bjorn Knollmann

Reviewing Editor: T Alexander Quinn

Transaction Report:

Dear Dr Delbridge,

Re: JP-RP-2024-286437 "Mechanical loading reveals an intrinsic cardiomyocyte stiffness contribution to diastolic dysfunction in murine cardiometabolic disease" by Johannes V Janssens, Antonia J.A. Raaijmakers, Parisa Koutsifeli, Kate L Weeks, James R Bell, Jennifer E Van Eyk, Claire L Curl, Kimberley M Mellor, and Lea M. D. Delbridge

Thank you for submitting your manuscript to The Journal of Physiology. It has been assessed by a Reviewing Editor and by 2 expert referees and we are pleased to tell you that it is potentially acceptable for publication following satisfactory major revision.

LANGUAGE EDITING AND SUPPORT FOR PUBLICATION: If you would like help with English language editing, or other article preparation support, Wiley Editing Services offers expert help, including English Language Editing, as well as translation, manuscript formatting, and figure formatting at www.wileyauthors.com/eoo/preparation. You can also find resources for Preparing Your Article for general guidance about writing and preparing your manuscript at www.wileyauthors.com/eoo/prepresources.

REVISION CHECKLIST:

We look forward to receiving your revised submission.

Yours sincerely,

Bjorn Knollmann
Senior Editor
The Journal of Physiology

REQUIRED ITEMS FOR REVISION

- Author photo and profile. First or joint first authors are asked to provide a short biography (no more than 100 words for one author or 150 words in total for joint first authors) and a portrait photograph. These should be uploaded and clearly labelled together in a Word document with the revised version of the manuscript. See Information for Authors for further details.
- You must start the Methods section with a paragraph headed Ethical Approval. A detailed explanation of journal policy and regulations on animal experimentation is given in Principles and standards for reporting animal experiments in The Journal of Physiology and Experimental Physiology by David Grundy J Physiol, 593: 2547-2549. doi:10.1113/JP270818). A checklist outlining these requirements and detailing the information that must be provided in the paper can be found at: <https://physoc.onlinelibrary.wiley.com/hub/animal-experiments>. Authors should confirm in their Methods section that their experiments were carried out according to the guidelines laid down by their institution's animal welfare committee, and conform to the principles and regulations as described in the Editorial by Grundy (2015), including an ethics approval reference number. The Methods section must contain a statement about access to food, water and housing, details of the anaesthetic regime: anaesthetic used, dose and route of administration, and method of killing the experimental animals.
- Please upload separate high-quality figure files via the submission form.
- Your paper contains Supporting Information of a type that we no longer publish, including supplementary tables and figures. Any information essential to an understanding of the paper must be included as part of the main manuscript and figures. The only Supporting Information that we publish are video and audio, 3D structures, program codes and large data files. Your revised paper will be returned to you if it does not adhere to our Supporting Information Guidelines.
- Papers must comply with the Statistics Policy: https://jp.msubmit.net/cgi-bin/main.plex?form_type=display_requirements#statistics.

In summary:

- If $n \leq 30$, all data points must be plotted in the figure in a way that reveals their range and distribution. A bar graph with data points overlaid, a box and whisker plot or a violin plot (preferably with data points included) are acceptable formats.
- If $n > 30$, then the entire raw dataset must be made available either as supporting information, or hosted on a not-for-profit repository, e.g. FigShare, with access details provided in the manuscript.
- 'n' clearly defined (e.g. x cells from y slices in z animals) in the Methods. Authors should be mindful of pseudoreplication.
- All relevant 'n' values must be clearly stated in the main text, figures and tables.
- The most appropriate summary statistic (e.g. mean or median and standard deviation) must be used. Standard Error of the

Mean (SEM) alone is not permitted.

- Exact p values must be stated. Authors must not use 'greater than' or 'less than'. Exact p values must be stated to three significant figures even when 'no statistical significance' is claimed.

- Please include an Abstract Figure file, as well as the Figure Legend text within the main article file. The Abstract Figure is a piece of artwork designed to give readers an immediate understanding of the research and should summarise the main conclusions. If possible, the image should be easily 'readable' from left to right or top to bottom. It should show the physiological relevance of the manuscript so readers can assess the importance and content of its findings. Abstract Figures should not merely recapitulate other figures in the manuscript. Please try to keep the diagram as simple as possible and without superfluous information that may distract from the main conclusion(s). Abstract Figures must be provided by authors no later than the revised manuscript stage and should be uploaded as a separate file during online submission labelled as File Type 'Abstract Figure'. Please also ensure that you include the figure legend in the main article file. All Abstract Figures should be created using BioRender. Authors should use The Journal's premium BioRender account to export high-resolution images. Details on how to use and access the premium account are included as part of this email.

- Please include a full title page as part of your main article (Word) file, which should contain the following: title, authors, affiliations, corresponding author name and contact details, keywords, and running title.

EDITOR COMMENTS

Reviewing Editor: Methods Details:

The method of euthanasia prior to heart isolation has not been stated.

Comments for Authors to ensure the paper complies with the Statistics Policy:

Values should be reported as mean +/- SD (not SEM).

Comments to the Author:

Your paper has been reviewed by two experts in the field, both of whom were enthusiastic, highlighting that the study was innovative, the experiments carefully conducted, and the data are convincing. They both also felt the manuscript is well written and presents novel and original findings that have the potential to be influential in the field. That said, the reviewers have some methodological and analytical concerns, and have identified important aspects that need clarification, as outline in their Comments for the Author. These should be addressed in a revision of the manuscript, along with a point-by-point response to the reviewers' concerns.

Senior Editor:

Comments for Authors to ensure the paper complies with the Statistics Policy:

as per reviewing editor

Comments to the Author:

I concur with the reviewing editor. A responsive revision will require substantial new experimental data to address the concerns raised in the review.

REFeree COMMENTS

Referee #1:

In this study, authors describe the effects of a high fat/ sugar diet (HFSD) on mouse cardiomyocytes passive mechanical properties and on calcium handling. A relationship between CM stiffness and in vivo dysfunction severity is observed. This study shows that a component of cardiac diastolic

dysfunction in cardiometabolic disease derives from intrinsic cardiomyocyte mechanical abnormality. This work also highlights that cell mechanical loading allows to reveal phenotypes that are not visible in unloaded conditions, the condition for most studies.

Data are very clearly presented and the manuscript is easy to read. Data quality is quite impressive to us (very challenging experiments and quite a low variability). We think this article will be of interest to the JP readership, it is original and describes important novel insights related to cardiomyocyte mechanics in a pathological context (high fat/sugar diet), but some aspects need to be clarified as indicated below.

Major revisions

Diastolic SL of almost all cells is below 1.8 and a significant number of cells have a SL even below 1.7 μm (Fig 2 B) which is often the lower limit for diastolic SL of healthy cells. Is there any explanation for this? When considering only cells having $\text{SL} > 1.7 \mu\text{m}$, are the results unchanged? Does diastolic SL (of unloaded cells) have any effects on the observations reported?

L138: stiffness and the Young's Modulus are not identical. Stiffness is a structural property, influenced by the geometry of the specimens as well as the material(s) of which it is comprised. Young's modulus is a material property, that is intrinsic to the material, and is not influenced by specimen geometry. As a way to simplify (as Biologists mainly will constitute the readership), the Young's Modulus is commonly referred to as a measure of sample stiffness but it should be clear in the manuscript that stiffness and the Young's Modulus are not equal.

Figure S1D: a strain of 5 is shown, what does it correspond to? Could authors give more explanation about the 'stretch constant'? Figure 4D shows nicely the segment length change which is very reproducible and of a maximum of about 30%. Interestingly SL changes much less, about 8-10%. Could authors discuss this difference? Is it the glue that is deforming, could the rods slide? Is it because the cells are only attached from the top? Could this difference explain the very large difference of 'stiffness' between the segment and the sarcomere? If yes, the sarcomere deformation is the most reliable by far and should be the only one considered (in addition, sarcomere's 'stiffness' values are closer to what one could expect from the literature).

Figure 4: Systolic force is shown for control cells and it would be useful to show it also for cardiomyocytes from HFSD mice. Is the Frank Starling gain index (<https://pubmed.ncbi.nlm.nih.gov/21494804/>) different between control and HFSD cells? From Figure 4A, it looks like the cardiomyocyte from HFSD produces more work compare to the control cell. In general, alterations (if any) of active forces should be described. We understand that the focus of the paper is on passive mechanics but recordings, Fig4 and 6 for example, show active mechanics/ calcium data and we find very difficult to ignore this aspect (data have been obtained, are shown and not quantified).

Figure 5: Ca^{2+} transient decay is shown, it would be informative to add complementary parameters: Ca^{2+} transient amplitude and speed of Ca^{2+} concentration increase as well as systolic and diastolic levels

Definition of the Young's Modulus. Interestingly the term 'Young's Modulus' is used only in the introduction and in the discussion (sometimes with inverted commas) but not in the result part. This raises the question of knowing whether data presented in Figure 4D and F show really the Young's Modulus. The Young's Modulus is stress/strain which is what is quantified in the figures, but it is usually acquired using ramp protocols (steady speed of force application) to induce the deformation, this is in part to reduce to a minimum the effect of viscosity on the measurement. Here a relatively long step protocol is used, in this condition the cells have time to relax partially, modifying the final result (lowering the Young's Modulus). Viscosity should be quantified and we are wondering whether faster stretch or/and ramp protocols have been obtained so the Young's Modulus could be calculated from these. At least this should be discussed.

Statistics: when sample size is low (below 15), verifying normal distribution is challenging and for this reason we suggest to use a non-parametric test instead of the t-test.

Minor revisions

Please add consistently space between the number and the unit (eg 276,)

L34-35: 'There are no specific treatments for diastolic dysfunction and therapies to manage symptoms have limited efficacy'. The structure of the sentence is incorrect, please rephrase.

L37: 'stiffness (stress/strain)', we would delete '(stress/strain)' or give a proper definition of stiffness

L49: 'With transition from 2-4Hz', we suggest to write 'from 2 to 4 Hz'.

L53-56: the conclusion of the abstract is vague (e.g.'cardiomyocyte mechanical abnormality'), we suggest to be more precise and give the exact effects that were observed. We think authors can be

Fig1: abbreviation of minutes is 'min' in the International System of Units not 'mins', units don't take a 's'. Same for 'sec' in figure 2.

early mitral inflow (E wave measurements): regarding the variability, probably not very meaningful to give a precision in the range of 0.1 mm/s.

L191: 'retrograde perfusion', the perfusion is retrograde only on a very short portion of the aorta (from the tip of the cannula to the coronary ostia) the direction of the flow in the coronaries is not retrograde, for this reason we suggest to not use this wording.

L283: 'there was increased early mitral inflow'. Sensus stricto, to be able to state that there is an increase, a time course should be visible like in A. If there is a single time point that has been observed we suggest to use 'higher' instead of 'increased'. Please check throughout the manuscript.

L287: 'No differences in systolic performance parameters': we suggest to say 'no significant differences'. Same L298, 300.

L427: 'A trend for an interaction effect ($p=0.055$) between diet and load was observed'. Please use the relevant statistical test to demonstrate whether there is a trend or not.

Figure S1C: unit for the x-axis is missing

Figure 4B: how is systolic stress calculated? Is max stress obtained from 1 transient or from many?

Referee #2:

Comments to authors:

General:

Diastolic dysfunction occurs when left ventricular stiffness increases and impairs the ability of the heart to relax and fill with blood. It occurs often in those with cardiometabolic conditions and can lead to heart failure with preserved ejection fraction (HFpEF), a condition which is increasing in the population. There is a lack of diagnostics and treatments for diastolic dysfunction, potentially because the underlying mechanisms are not well understood. In this innovative study, the authors investigate the role that intrinsic ventricular myocyte stiffness plays in diastolic dysfunction in a high fat diet-induced model of diastolic dysfunction. Here, the authors provide convincing evidence that intrinsic cardiomyocyte stiffness contributes to diastolic dysfunction in a mouse model of cardiometabolic disease. Critically, they show that stretch-dependent augmentation of myofilament responses to calcium in diastole contribute to the increased cardiomyocyte stiffness and diastolic dysfunction in mice with cardiometabolic disease.

Specific Comments:

The model of cardiometabolic disease that the authors use to induce diastolic dysfunction is introduced in the methods and justified in the discussion. Until the discussion it is not clear if this has been demonstrated previously and the information comes very late in the manuscript. The model should be introduced, even briefly, in the introduction to clarify that this is an established model of diastolic dysfunction. In fact, the information presented on lines 502 to 517 would seem to be a much better fit in the introduction to establish the model and highlight why the author's approach is an advance over what has previously been shown with this model.

In Figure 3 B and C, the authors have pooled the two diet conditions to illustrate the impact of loading on shortening extent and lengthening rate. It is hard to appreciate the individual data points in the figure and it is not entirely clear why these data have been pooled. It does look like the effects on the control cells might be smaller than the effects on the HFD cells. Is this the case? It would be helpful to present these two figure panels as unpooled data or to add this to the data supplement, with a justification for pooling the data in the text.

The authors present strong evidence for diastolic dysfunction in the in vivo experiments and in the cell experiments. However, they have not reported IVRT measurements, which are often used to assess diastolic dysfunction in mice (PMID: 29351456). There is some evidence that this might be a better measure of diastolic dysfunction certainly than E/A ratios (PMID: 29055654), although I recognize that the authors do not rely on these measures here. It would be interesting to include these measurements if available, regardless of whether they indicate diastolic dysfunction, and discuss this in the revised manuscript.

The authors have only investigated male mice, and this is a limitation to the study, as acknowledged by the authors in the discussion. It would be important to note that this study uses male mice in the title and abstract in revising this work.

It seems redundant to note the specific figures and tables, by number, in the discussion.

END OF COMMENTS

Mechanical loading reveals an intrinsic an intrinsic cardiomyocyte stiffness contribution to diastolic dysfunction in murine cardiometabolic disease

Johannes V. Janssens, Antonia J.A. Raaijmakers, Parisa Koutsifeli, Kate L. Weeks, James R. Bell, Jennifer E. Van Eyk, Claire L. Curl, Kimberley M. Mellor, Lea M.D. Delbridge

RESPONSE TO REVIEWERS

We appreciate the feedback provided by the Reviewers and Editors, who have generously taken time to consider the content and offer strategies for revision. The suggestions for inclusion/ expansion of additional data and commentary relating to the execution of the study are very welcome. We are encouraged by comments from the reviewers on the study highlighting that “the data quality is quite impressive”, “it describes important novel insights related to cardiomyocyte mechanics” and “is innovative”. All matters raised by the Reviewers have been addressed as detailed below. New data relating to systolic stress (ie Frank-Starling Gain) and systolic Ca²⁺ parameters are now included within the manuscript as one new figure panel (**Fig. 6B**) and two additional tables (**Tables. 2&3**). New commentary is included within the manuscript relating to: distinguishing stiffness from Young’s Modulus; considering appropriate non-loaded sarcomere lengths for inclusion; interpreting results from systolic data; discussing echocardiographic parameters of diastolic dysfunction (7 new reference citations have been added – total citation count = 78).

Reviewer comments are incorporated verbatim (**bold**) below and modifications to the manuscript have been identified *using italics* with reference to the revised manuscript pagination. In the uploaded revised manuscript, inserted text portions and changes relating to Reviewer comments are shown in **yellow highlight** to assist with Reviewer/Editor tracking and checking.

Reviewing Editor (item #'s inserted to assist cross-referencing in response)

The method of euthanasia prior to heart isolation has not been stated.

New text inserted: “Prior to cardiomyocyte isolation animals were anaesthetised (sodium pentobarbital 70 mg/kg i.p) and hearts excised.”

Page 7, Lines 203-204

Values should be reported as mean +/- SD (not SEM).

Standard deviations have now been included in relevant figures. In panels where both x and y-axis error is depicted we have retained the +/- SEM format (5B, C, E, & 7E) to assist reader evaluation of the data. These data could alternatively be shown without any error representation. We will be happy to take editor advice on this.

Reviewer 1 (item #'s inserted to assist cross-referencing in response)

Major Revisions

Diastolic SL of almost all cells is below 1.8 and a significant number of cells have a SL even below 1.7 µm (Fig 2 B) which is often the lower limit for diastolic SL of healthy cells. Is there any explanation for this?

New text inserted: “A range of non-loaded cardiomyocyte mean diastolic sarcomere lengths are reported in the literature. When cells are evaluated in vitro and not constrained by local structural elements sarcomere lengths <1.8µm are commonly observed and consistent with our findings (Lim et al., 2000; Bub et al., 2010). In vitro experimental conditions can also contribute to variation in diastolic sarcomere

lengths (e.g. pacing frequency, temperature and buffer tonicity) (Roos et al., 1982; Chung & Campbell, 2013; Khokhlova et al., 2022).”

Citations inserted: PMID: 20228259, PMID: 35444559, PMID: 35163643, PMID: 6277204, PMID: 24400159
Page 18, Lines 563 - 568

When considering only cells having SL>1.7 µm, are the results unchanged? Does diastolic SL (of unloaded cells) have any effects on the observations reported?

When cardiomyocytes of SL >1.7µm are selectively analysed (data from Fig. 2) a faster maximal rate of sarcomere lengthening (CTRL:1.10±0.98, HFSD: 1.88±0.30 µm/s) and a reduced tau (CTRL:0.177±0.081, HFSD: 0.133±0.060 µm/s) are computed for HFSD unloaded cardiomyocytes relative to control. Thus, the results of this selective analysis excluding data based on sarcomere length of less than 1.7µm are consistent with (and indeed reinforce) the overall conclusions described in the manuscript. However, based on the considerations set out in the response paragraph immediately above, the use of a diastolic SL threshold of 1.7µm is difficult to justify for our data and may suppress biological variability. For this reason we have retained the full data set in the manuscript.

L138: stiffness and the Young's Modulus are not identical. Stiffness is a structural property, influenced by the geometry of the specimens as well as the material(s) of which it is comprised. Young's modulus is a material property, that is intrinsic to the material, and is not influenced by specimen geometry. As a way to simplify (as Biologists mainly will constitute the readership), the Young's Modulus is commonly referred to as a measure of sample stiffness but it should be clear in the manuscript that stiffness and the Young's Modulus are not equal.

This is a helpful distinction to make for readers. We have incorporated the concepts highlighted by the reviewer in a revised paragraph in the Introduction which now reads: “*Stiffness is defined as the extent to which an object resists deformation. Stiffness is influenced by the geometry and the intrinsic material properties of an object. In the cardiac literature and in this study, stress/strain relations are calculated to assess cardiomyocyte resistance to stretch. Stress is defined as force per cross-sectional area, and strain is the application of a normalized stretch or deformation. The ratio of stress/strain defines the cardiomyocyte resistance to stretch independent of geometry and is known as ‘Young’s Modulus’. While stiffness and Young’s Modulus are not equivalent, the term ‘stiffness’ is commonly used (even though Young’s Modulus is the correct formal term).*”

Pg 5, Lines 140-147

Figure S1D: a strain of 5 is shown, what does it correspond to?

The figure axis has been labelled more clearly as “%initial segment length”. The legend has been edited similarly.

Figure 4 (previously Fig. S1)

Could authors give more explanation about the 'stretch constant'?

Explanation is provided: “*For each myocyte a Calibration Constant was calculated as: myocyte segment length change per piezo-motor mV / sarcomere count in segment [where myocyte sarcomere count = myocyte segment length (µm) / average diastolic sarcomere length (µm)].*”

Pg 8, Lines 259-261

Figure 4D shows nicely the segment length change which is very reproducible and of a maximum of about 30%. Interestingly SL changes much less, about 8-10%. Could authors discuss this difference? Is it the glue that is deforming, could the rods slide? Is it because the cells are only attached from the top? Could this difference explain the very large difference of 'stiffness' between the segment and the sarcomere? If yes, the sarcomere deformation is the most reliable by far and should be the

only one considered (in addition, sarcomere's 'stiffness' values are closer to what one could expect from the literature).

Fig S1C lower plot (now Figure 4C) shows that glass fibre position changes are linear for the total range of stretch steps indicating the rigidity (ie non 'sliding') of the glass fibres (we have interpreted the term 'rod' to mean fibre). Sarcomere deformation measurement maybe limited at least partly by the issue of top surface adhesion as described by the reviewer and this may be a reason why the segment measures were more reproducible. An additional limitation of mean sarcomere measurement is that the extent of sarcomere length change during stretch is not uniform within the segment. This particular observation has reported elsewhere recently (PMID: 37401464).

Text has been inserted in the limitations section to highlight these issues: *"An additional aspect of our pragmatic approach has been to evaluate cardiomyocyte segment stretches as average sarcomere stretch methods can be challenging to track optically."*

Page 20, Lines 651-653

Figure 4: Systolic force is shown for control cells and it would be useful to show it also for cardiomyocytes from HFSD mice.

These data have been added to panel B in the existing **Figure 5** (previously **Figure 4**).

Is the Frank Starling gain index (<https://pubmed.ncbi.nlm.nih.gov/21494804/>) different between control and HFSD cells? From Figure 4A, it looks like the cardiomyocyte from HFSD produces more work compare to the control cell.

Data regarding the Frank Starling Gain Index have been added to the manuscript in a new table (Table 2). No significant difference was observed in FSG index between CTRL and HFSD.

Text inserted into results: *"There was no significant difference in 'Frank Starling Gain index' (Table. 2, CTRL: 2.40 ± 1.06 , HFSD: 2.77 ± 1.45 , (mN/mm²) / μ m) or the ratio of systolic to diastolic stress segment length relations (Table. 2, CTRL: 1.62 ± 0.25 , HFSD: 1.73 ± 0.50 , (mN/mm²) / % length) indicating that cardiomyocyte length-dependent activation is preserved (Bollensdorff et al., 2011; Najafi et al., 2016)."*

Pg 13, Lines 418 – 421

Citations added: PMID: 21494804, PMID: 26825555

In general, alterations (if any) of active forces should be described. We understand that the focus of the paper is on passive mechanics but recordings, Fig4 and 6 for example, show active mechanics/ calcium data and we find very difficult to ignore this aspect (data have been obtained, are shown and not quantified).

Two new tables have been incorporated (Tables 2&3) including additional stress and Ca²⁺ systolic parameters as suggested. Text has been inserted into the methods to describe the additional parameters that have been included in Tables 2&3.

Text inserted into Methods: *"...systolic : diastolic stress-sarcomere relation (Frank Starling Gain Index (Bollensdorff et al., 2011)), systolic : diastolic stress-length relation, systolic stress amplitude (mN/mm²)..."*

Pg 9, Lines 273-275

Text inserted into Results: *"No treatment or interaction effects were observed for systolic Ca²⁺, Ca²⁺ amplitude, or time constant of Ca²⁺ transient decay (Table. 3)."*

Pg 14-15, Lines 463-464

Figure 5: Ca²⁺ transient decay is shown, it would be informative to add complementary parameters: Ca²⁺ transient amplitude and speed of Ca²⁺ concentration increase as well as systolic and diastolic levels

An additional panel (**Fig. 6B**) has now been included in **Figure 6** (previously **Figure 5**) to provide information about systolic Ca²⁺ time to peak for each condition investigated. Systolic and diastolic Ca²⁺

levels have been provided for non-loaded cardiomyocytes in **Figure 2** and for loaded cardiomyocytes in **Table 3**.

New text inserted into Results: “Loading prolonged time to peak Ca^{2+} transient in both CTRL and HFSD cardiomyocytes (**Fig. 6B**. CTRL: non-loaded 1.00 ± 0.25 ; loaded 1.49 ± 0.38 ; stretched 1.46 ± 0.44 , HFSD: non-loaded 1.00 ± 0.41 ; loaded 1.36 ± 0.41 ; stretched 1.21 ± 0.34 fold change).”

Pg 14, Lines 444-447

Definition of the Young's Modulus. Interestingly the term 'Young's Modulus' is used only in the introduction and in the discussion (sometimes with inverted commas) but not in the result part. This raises the question of knowing whether data presented in Figure 4D and F show really the Young's Modulus. The Young's Modulus is stress/strain which is what is quantified in the figures, but it is usually acquired using ramp protocols (steady speed of force application) to induce the deformation, this is in part to reduce to a minimum the effect of viscosity on the measurement. Here a relatively long step protocol is used, in this condition the cells have time to relax partially, modifying the final result (lowering the Young's Modulus). Viscosity should be quantified and we are wondering whether faster stretch or/and ramp protocols have been obtained so the Young's Modulus could be calculated from these. At least this should be discussed.

Text has been included in the Introduction regarding the usage of the terms stiffness and Young's Modulus to provide the context for the usage of the term stiffness in the Results.

In the limitations section we have expanded the description of the impact of step protocols vs ramp protocols on measurement of viscous contribution to stiffness quantification: “Our protocol involves the application of cumulative ramp steps with about 20% non-loaded-to-loaded myocyte conversion success rate.The two approaches have not been benchmarked, although it can be predicted that the viscous components of stiffness are likely underestimated when stretches are slow and relatively small such as in the current study (Caporizzo & Prosser, 2021).”

Pg 20, Lines 645-650

Statistics: when sample size is low (below 15), verifying normal distribution is challenging and for this reason we suggest to use a non-parametric test instead of the t-test.

As per standard guidelines we have tested for normality violation in our statistical analyses before proceeding to parametric statistical analysis.

Minor revisions

Please add consistently space between the number and the unit (eg 276,)

Done

L34-35: 'There are no specific treatments for diastolic dysfunction and therapies to manage symptoms have limited efficacy'. The structure of the sentence is incorrect, please rephrase.

Punctuation added.

L37: 'stiffness (stress/strain)', we would delete '(stress/strain)' or give a proper definition of stiffness

Text deleted: '(Stress/ strain)'

Pg 2, Line 37

L49: 'With transition from 2-4Hz', we suggest to write 'from 2 to 4 Hz'.

'2-4 Hz' has been removed

L53-56: the conclusion of the abstract is vague (e.g.'cardiomyocyte mechanical abnormality'), we suggest to be more precise and give the exact effects that were observed. We think authors can be...(this reviewer comment appears to be incomplete)

The final two sentences of the abstract have been edited: *“Collectively, these findings demonstrate that a component of cardiac diastolic dysfunction in cardiometabolic disease is derived from cardiomyocyte stiffness. Differential responses to load, stretch and pacing suggest that a previously undescribed alteration in myofilament-Ca²⁺ interaction contributes to intrinsic cardiomyocyte stiffness in cardiometabolic disease.”*

Pg 2, Lines 52-55

Fig1: abbreviation of minutes is 'min' in the International System of Units not 'mins', units don't take a 's'. Same for 'sec' in figure 2.

Done

early mitral inflow (E wave measurements): regarding the variability, probably not very meaningful to give a precision in the range of 0.1 mm/s.

Number of significant figures adjusted

Pg 10, Line 296 & Table 1

L191: 'retrograde perfusion', the perfusion is retrograde only on a very short portion of the aorta (from the tip of the cannula to the coronary ostia) the direction of the flow in the coronaries is not retrograde, for this reason we suggest to not use this wording.

Text removed: *“subjected to retrograde”*

Pg 7, Line 202

L283: 'there was increased early mitral inflow'. Sensu stricto, to be able to state that there is an increase, a time course should be visible like in A. If there is a single time point that has been observed we suggest to use 'higher' instead of 'increased'. Please check throughout the manuscript.

Text revised: *“there was higher early mitral inflow”*

Pg 10, Line 296

L287: 'No differences in systolic performance parameters': we suggest to say 'no significant differences'. Same L298, 300.

The words significant / significantly were inserted

Pg 10, Lines 300, 309 & 312

L427: 'A trend for an interaction effect (p=0.055) between diet and load was observed'. Please use the relevant statistical test to demonstrate whether there is a trend or not.

Text removed and comparison no longer shown in Figure 5A

Pg 14, Line 447

Figure S1C: unit for the x-axis is missing

Units added (Now Fig. 4C)

Figure 4B: how is systolic stress calculated? Is max stress obtained from 1 transient or from many?

Text inserted into methods: *“An average of at least 10 twitches or transients were used to calculate sarcomere, Ca²⁺ and force parameters at a given stretch length.”*

Pg 9, Line 270-271

Reviewer 2 (item #'s inserted to assist cross-referencing in response)

The model of cardiometabolic disease that the authors use to induce diastolic dysfunction is introduced in the methods and justified in the discussion. Until the discussion it is not clear if this has been demonstrated previously and the information comes very late in the manuscript. The model should be introduced, even briefly, in the introduction to clarify that this is an established model of diastolic dysfunction. In fact, the information presented on lines 502 to 517 would seem to be a much better fit in the introduction to establish the model and highlight why the author's approach is an advance over what has previously been shown with this model.

Text has been inserted into the Introduction: *“The experimental induction of these phenotypes using a high/fat sugar diet feeding protocol has been shown by us and others to reliably produce diastolic dysfunction in rodents (Pulinilkunnil et al., 2014; Dia et al., 2020; Sowers et al., 2020; Wingard et al., 2021; Daniels et al., 2022)”*

Pg 4, lines 90-92

In Figure 3 B and C, the authors have pooled the two diet conditions to illustrate the impact of loading on shortening extent and lengthening rate. It is hard to appreciate the individual data points in the figure and it is not entirely clear why these data have been pooled. It does look like the effects on the control cells might be smaller than the effects on the HFD cells. Is this the case? It would be helpful to present these two figure panels as unpooled data or to add this to the data supplement, with a justification for pooling the data in the text.

The goal in Figure B & C was to investigate the general effect of loading on all cardiomyocytes of any treatment group. The goal was not specifically to compare the groups in these panels. To address this reviewer question a statistical analysis (2-way ANOVA) showed there is no significant load x diet interaction effect between CTRL and HFSD groups in panels 3B & C. The information about treatment effects was already incorporated in Fig. 3E&F.

Text inserted into results to clarify purpose of the data presentation approach: *“In the first instance, data from all myocytes were pooled to investigate the general effects of load regardless of treatment group.”*

Pg 11, Line 338-339

The authors present strong evidence for diastolic dysfunction in the in vivo experiments and in the cell experiments. However, they have not reported IVRT measurements, which are often used to assess diastolic dysfunction in mice (PMID: 29351456). There is some evidence that this might be a better measure of diastolic dysfunction certainly than E/A ratios (PMID: 29055654), although I recognize that the authors do not rely on these measures here. It would be interesting to include these measurements if available, regardless of whether they indicate diastolic dysfunction, and discuss this in the revised manuscript.

We have recently published an analysis of IVRT in male rodents fed the same diet for similar duration and shown slight shortening of IVRT (1.7ms, PMID: 37524893). It is challenging to interpret the meaning of such a small effect, and in the context of the wider literature where there may be no change or prolongation of IVRT. For these reasons we have not pursued IVRT analysis in this data set.

Text inserted into the Discussion: *“High fat/sugar diet feeding (i.e. ‘western diet’) has previously been shown experimentally to induce diastolic dysfunction in rodents. Elevated E/e’ is a remarkably consistent echocardiographic feature of high fat/sugar diet fed mouse myocardial performance while E/A and IVRT may be elevated, reduced, or unchanged relative to control counterparts (Pulinilkunnil et al., 2014; Dia et al., 2020; Sowers et al., 2020; Wingard et al., 2021; Daniels et al., 2022).”*

Pg 17, Lines 524-527

The authors have only investigated male mice, and this is a limitation to the study, as acknowledged by the authors in the discussion. It would be important to note that this study uses male mice in the title and abstract in revising this work.

Information about animal sex has been included in the abstract. We suggest that inclusion of the sex of the animals studied in the title may infer that the work relates to a sex-specific finding – which is not the case. We defer to editorial decision regarding the inclusion of animal sex in the title.

Text inserted into Abstract: *“Male mice fed a High Fat/Sugar Diet (HFSD vs control)...”*

Pg 2, Line 40

It seems redundant to note the specific figures and tables, by number, in the discussion.

Reference to specific figure and table numbers removed from the discussion.

Pgs 16-17

Dear Dr Delbridge,

Re: JP-RP-2024-286437R1 "Mechanical loading reveals an intrinsic cardiomyocyte stiffness contribution to diastolic dysfunction in murine cardiometabolic disease" by Johannes V Janssens, Antonia J.A. Raaijmakers, Parisa Koutsifeli, Kate L Weeks, James R Bell, Jennifer E Van Eyk, Claire L Curl, Kimberley M Mellor, and Lea M. D. Delbridge

Thank you for submitting your manuscript to The Journal of Physiology. It has been assessed by a Reviewing Editor and by 2 expert referees and we are pleased to tell you that it is acceptable for publication following satisfactory revision.

REVISION CHECKLIST:

We look forward to receiving your revised submission.

Yours sincerely,

Bjorn Knollmann
Senior Editor
The Journal of Physiology

REQUIRED ITEMS FOR REVISION

- Please upload separate high-quality figure files via the submission form.
- Your paper contains Supporting Information of a type that we no longer publish, including supplementary tables and figures. Any information essential to an understanding of the paper must be included as part of the main manuscript and figures. The only Supporting Information that we publish are video and audio, 3D structures, program codes and large data files. Your revised paper will be returned to you if it does not adhere to our Supporting Information Guidelines.
- Papers must comply with the Statistics Policy: https://jp.msubmit.net/cgi-bin/main.plex?form_type=display_requirements#statistics.

In summary:

- If $n \leq 30$, all data points must be plotted in the figure in a way that reveals their range and distribution. A bar graph with data points overlaid, a box and whisker plot or a violin plot (preferably with data points included) are acceptable formats.
- If $n > 30$, then the entire raw dataset must be made available either as supporting information, or hosted on a not-for-profit repository, e.g. FigShare, with access details provided in the manuscript.
- 'n' clearly defined (e.g. x cells from y slices in z animals) in the Methods. Authors should be mindful of pseudoreplication.
- All relevant 'n' values must be clearly stated in the main text, figures and tables.
- The most appropriate summary statistic (e.g. mean or median and standard deviation) must be used. Standard Error of the Mean (SEM) alone is not permitted.
- Exact p values must be stated. Authors must not use 'greater than' or 'less than'. Exact p values must be stated to three significant figures even when 'no statistical significance' is claimed.

EDITOR COMMENTS

Reviewing Editor:

Please include details regarding: (i) the origin and source of animals and (ii) their access to food and water in the final version.

Senior Editor:

I concur with the reviewing editor. Please address the animal concerns per Journal guidelines.

REFEREE COMMENTS

Referee #1:

The authors properly addressed all our comments.

Referee #2:

The authors have done a good job in responding to my concerns, except they have not addressed the missing animal husbandry information.

END OF COMMENTS

Mechanical loading reveals an intrinsic an intrinsic cardiomyocyte stiffness contribution to diastolic dysfunction in murine cardiometabolic disease

Johannes V. Janssens, Antonia J.A. Raaijmakers, Parisa Koutsifeli, Kate L. Weeks, James R. Bell, Jennifer E. Van Eyk, Claire L. Curl, Kimberley M. Mellor, Lea M.D. Delbridge

RESPONSE TO REVIEWERS – REVISION 2

EDITOR COMMENTS

Reviewing Editor: Please include details regarding: (i) the origin and source of animals and (ii) their access to food and water in the final version.

Senior Editor: I concur with the reviewing editor. Please address the animal concerns per Journal guidelines.

These details are now included in the revised manuscript. Apologies for this oversight.

REFEREE COMMENTS

Referee #1: The authors properly addressed all our comments.

Referee #2: The authors have done a good job in responding to my concerns, except they have not addressed the missing animal husbandry information.

Apologies, this information is now provided in the revised version.

Dear Professor Delbridge,

Re: JP-RP-2024-286437R2 "Mechanical loading reveals an intrinsic cardiomyocyte stiffness contribution to diastolic dysfunction in murine cardiometabolic disease" by Johannes V Janssens, Antonia J.A. Raaijmakers, Parisa Koutsifeli, Kate L Weeks, James R Bell, Jennifer E Van Eyk, Claire L Curl, Kimberley M Mellor, and Lea M. D. Delbridge

We are pleased to tell you that your paper has been accepted for publication in The Journal of Physiology.

Yours sincerely,

Bjorn Knollmann
Senior Editor
The Journal of Physiology

If you would like to receive our 'Research Roundup', a monthly newsletter highlighting the cutting-edge research published in The Physiological Society's family of journals (The Journal of Physiology, Experimental Physiology, Physiological Reports, The Journal of Nutritional Physiology and The Journal of Precision Medicine: Health and Disease), please click this link, fill in your name and email address and select 'Research Roundup':

<https://www.physoc.org/journals-and-media/membernews>

- You can help your research get the attention it deserves! Check out Wiley's free Promotion Guide for best-practice recommendations for promoting your work at: www.wileyauthors.com/eo/guide. You can learn more about Wiley Editing Services which offers professional video, design, and writing services to create shareable video abstracts, infographics, conference posters, lay summaries, and research news stories for your research at: www.wileyauthors.com/eo/promotion.

The Corresponding Author will receive an email from Wiley with details on how to register or log-in to Wiley Authors Services where you will be able to place an order

EDITOR COMMENTS

The MS is now acceptable. Very nice work!